# DISCRIMINATING IMAGE REPRESENTATIONS WITH PRINCIPAL DISTORTIONS

**Jenelle Feather**[*1]  **David Lipshutz**[*1,2]  **Sarah E. Harvey**[1]

**Alex H. Williams**[1,3]  **Eero P. Simoncelli**[1,3]

[1]Center for Computational Neuroscience, Flatiron Institute, Simons Foundation
[2]Department of Neuroscience, Baylor College of Medicine
[3]Center for Neural Science, New York University

## ABSTRACT

Image representations (artificial or biological) are often compared in terms of their global geometric structure; however, representations with similar global structure can have strikingly different local geometries. Here, we propose a framework for comparing a set of image representations in terms of their local geometries. We quantify the local geometry of a representation using the Fisher information matrix, a standard statistical tool for characterizing the sensitivity to local stimulus distortions, and use this as a substrate for a metric on the local geometry in the vicinity of a base image. This metric may then be used to optimally differentiate a set of models, by finding a pair of "principal distortions" that maximize the variance of the models under this metric. As an example, we use this framework to compare a set of simple models of the early visual system, identifying a novel set of image distortions that allow immediate comparison of the models by visual inspection. In a second example, we apply our method to a set of deep neural network models and reveal differences in the local geometry that arise due to architecture and training types. These examples demonstrate how our framework can be used to probe for informative differences in local sensitivities between complex models, and suggest how it could be used to compare model representations with human perception.

## 1 INTRODUCTION

Biological and artificial neural networks transform sensory stimuli into high-dimensional internal representations that support downstream tasks, and these representations are often described in terms of their neural population geometry (Chung & Abbott, 2021). This idea has led to a multitude of proposed measures of representational similarity (Kriegeskorte et al., 2008; Yamins & DiCarlo, 2016; Kornblith et al., 2019; Williams et al., 2021; Klabunde et al., 2023), which have been used to compare representations within a computational model to the representations within a brain. However, despite differing in architectures and training procedures, many computational models of perceptual or neural responses are equally performant on these representational similarity measures (Schrimpf et al., 2018; Tuckute et al., 2023; Conwell et al., 2024). Are these models functionally interchangeable, or are the datasets and methods that are used to test them simply insufficient to reveal their differences?

Often the similarity between two representations is quantified by measuring alignment of the representations over a set of natural stimuli that are relatively far apart in stimulus space. In this way, these measures capture notions of *global* geometric similarity between representations. However, systems with similar global structure can have strikingly different *local* geometries. These local geometries can be investigated by measuring the sensitivity of a system to small image distortions. For example, Szegedy et al. (2013) found image distortions that were imperceptible to humans but

---

[*]Equal contribution, ordered alphabetically. Correspondence: `jfeather@flatironinstitute.org`, `david.lipshutz@bcm.edu`.

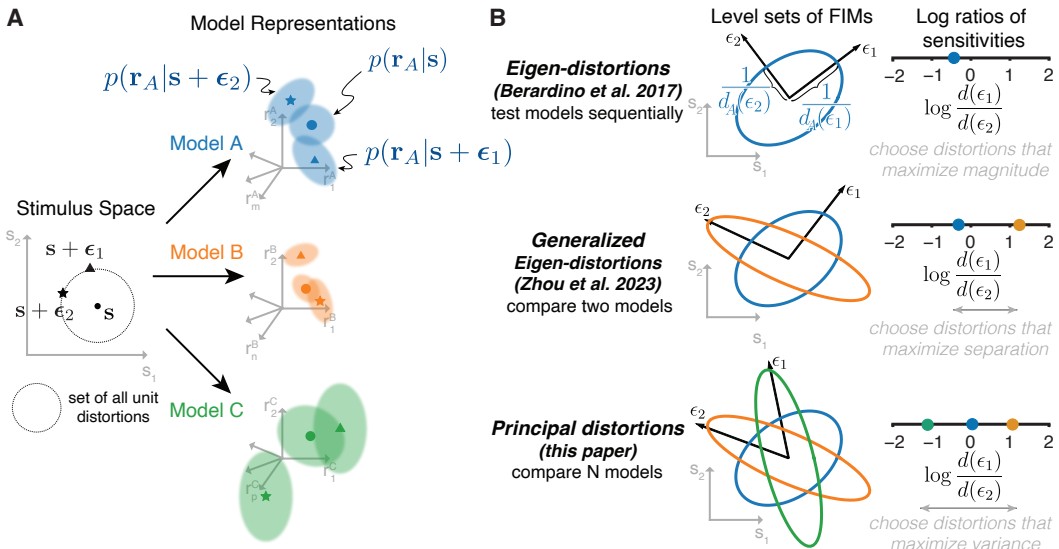

Figure 1: Comparing the local geometry of image representations. **A)** Each model is assumed to map stimuli to stochastic responses in a representation space—deterministic models can be investigated by assuming additive Gaussian response noise. For example, Model A maps the stimulus $\mathbf{s}$ (solid black circle ●) to a conditional density $p(\mathbf{r}_A|\mathbf{s})$ in Model A's representation space (solid blue circle ● and surrounding transparent blue ellipse). **B)** Model sensitivity at the base image $\mathbf{s}$ to local distortions can be mapped back to the stimulus domain via the model's positive semidefinite FIM. In the top panel "Eigen-distortions (Berardino et al., 2017)", the blue ellipse represents the unit level set $\{\mathbf{v} : d_A(\mathbf{v}) = 1\}$ of the norm induced by Model A's FIM $\mathbf{I}_A$, which is the set of distortions of the base stimulus $\mathbf{s}$ that appear equally distorted according to Model A's representation. The eigenvectors of the FIM ($\boldsymbol{\epsilon}_1, \boldsymbol{\epsilon}_2$) can equivalently be interpreted as the distortions that maximize the magnitude of the log ratio of the model's sensitivities (solid blue circle ● on the number line). In the middle panel "Generalized Eigen-distortions (Zhou et al., 2023)", the blue ellipse is copied from the top panel and the orange ellipse is the level set of Model B's FIM $\mathbf{I}_B$. The generalized eigenvectors of $\mathbf{I}_A$ and $\mathbf{I}_B$ ($\boldsymbol{\epsilon}_1, \boldsymbol{\epsilon}_2$) can equivalently be interpreted as the vectors that maximize the difference between the log ratios of the models' sensitivities. In the bottom panel "Principal distortions (this paper)", the blue and orange ellipses are as in the above panels and the green ellipse represents the level set of Model C's FIM $\mathbf{I}_C$. Here, we see two stimulus distortions ($\boldsymbol{\epsilon}_1, \boldsymbol{\epsilon}_2$) that maximize the variance of the log ratios of model sensitivities.

led artificial neural networks to misclassify images, which motivated methods for training artificial neural networks so as to minimize their susceptibility to these adversarial examples (Goodfellow et al., 2014; Madry, 2017). These observations suggest a need for metrics that compare the local geometries of image representations and, in particular, highlight the differences between systems even when global structure seems similar. Furthermore, such metrics may also aid in the development of network interpretability and explainability tools.

How can we quantify and compare the local geometry of different image representations? A brute-force comparison clearly is prohibitive: the space of images is extremely high-dimensional, and the set of potential distortions equally high-dimensional. Estimating the local geometry of representations over a moderately dense sampling of this full set of possible distortions is impractical, and estimating human sensitivity to such a set is essentially impossible. As such, it is worthwhile to develop a method for judicious selection of stimulus distortions that can be used when comparing a set of models.

We take inspiration from Zhou et al. (2023). For a pair of models and a base image, they synthesize distortions along which the two models' sensitivities maximally disagree. This bears conceptual similarity to other methods that construct stimuli to optimally distinguish a pair of models (Wang & Simoncelli, 2008; Golan et al., 2020) or cluster neurons by cell type (Burg et al., 2024), and builds on earlier work that examined "eigen-distortions" along which individual models are maxi-

mally/minimally sensitive (Berardino et al., 2017). Specifically, Zhou et al. measure the local sensitivity of a model in terms of its Fisher Information matrix (FIM, Fisher, 1925), a classical tool from statistical estimation theory, and choose the pair of "generalized eigen-distortions" that maximize/minimize the ratio of the two models' sensitivities. Once these image distortions have been computed, they may be added in varying amounts to a base image to determine the level at which they become visible to a human. These measured human sensitivities can then be compared to those of the models, with the goal of identifying which model is better aligned with the local geometry of the human visual system. The distortions can also be used as model interpretability tools. For instance, visualizations of the distortions may offer insight into how complex models systematically differ, providing practitioners with better understanding of model vulnerabilities. However, there is no principled method for selecting image distortions for comparing more than two models.

Here we define a novel metric for comparing model representations in terms of their relative sensitivities to image distortions. We then use this metric to generate a pair of distortions that maximize the *variance* across two or more models under this metric. In analogy with principal component analysis, our method can be viewed as a dimensionality reduction technique that preserves as much of the variability in the local representational geometry as possible. As such, we refer to these as the "principal distortions" of the set of models.

We apply our method to a nested set of hand-crafted models of the early visual system to identify distortions that differentiate these models and can potentially be used to evaluate how well these models predict human visual sensitivities. We then apply our method to a set of visual deep neural networks (DNNs) with varying architectures and training procedures. We find distortions that allow for visualization of differences in the sensitivities between layers of the networks and neural network architectures. We further explore differences between standard ImageNet trained networks and their shape-bias enhanced counterparts, and between standard networks and their adversarially-trained counterparts. In all cases, we illustrate how the method generates novel distortions that highlight differences between models. Portions of this work were initially presented in (Lipshutz* et al., 2024).

## 2 PROBLEM STATEMENT AND EXISTING METHODS

Given a collection of image representations, our goal is to develop a method for comparing the local geometries of these representations in the vicinity of some base input image. In this section, we define the local geometry of an image representation in terms of the FIM and review existing methods for selecting image distortions based on model FIMs.

### 2.1 LOCAL INFORMATION GEOMETRY OF STOCHASTIC IMAGE REPRESENTATIONS

We assume that each image representation has an associated conditional density $p(\mathbf{r}|\mathbf{s})$, where $\mathbf{s}$ is a $K$-dimensional vector of image pixels and $\mathbf{r}$ is a vector of stochastic responses (e.g., biological neuronal firing rates or deterministic model activations with additive response noise). Fig. 1A depicts a two-dimensional stimulus space and three models mapping stimuli $\mathbf{s}$, $\mathbf{s} + \boldsymbol{\epsilon}_1$, and $\mathbf{s} + \boldsymbol{\epsilon}_2$ to conditional densities. Note that the dimensions of the responses may vary across representations.

The sensitivity of the representation to a small distortion $\boldsymbol{\epsilon}$ depends on the overlap between the conditional distributions $p(\mathbf{r}|\mathbf{s})$ and $p(\mathbf{r}|\mathbf{s}+\boldsymbol{\epsilon})$, with less overlap corresponding to higher sensitivity (Green & Swets, 1966). This sensitivity can be precisely quantified in terms of the Fisher-Rao metric (Rao, 1945; Amari, 2016), a Riemannian metric on the stimulus space (Fig. 1B) that is defined in terms of the positive semi-definite FIM (Fisher, 1925):

$$\boldsymbol{I}(\mathbf{s}) := \mathbb{E}_{\mathbf{r} \sim p(\mathbf{r}|\mathbf{s})} \left[ \nabla_{\mathbf{s}} \log p(\mathbf{r}|\mathbf{s}) \nabla_{\mathbf{s}} \log p(\mathbf{r}|\mathbf{s})^{\top} \right],$$

where $\nabla_{\mathbf{s}} \log p(\mathbf{r}|\mathbf{s})$ denotes the gradient of $\log p(\mathbf{r}|\mathbf{s})$ with respect to $\mathbf{s}$. The FIM is a standard tool in statistical estimation theory that locally approximates the expected log-likelihood ratio (or KL divergence) between the conditional distributions $p(\mathbf{r}|\mathbf{s})$ and $p(\mathbf{r}|\mathbf{s}+\boldsymbol{\epsilon})$, and provides a lower bound on the variance of any unbiased estimator of $\mathbf{s}$ (Cramér, 1946; Rao, 1945). The FIM has also been used to link neural representations to perceptual discrimination (Paradiso, 1988; Seung & Sompolinsky, 1993; Brunel & Nadal, 1998; Averbeck & Lee, 2006; Seriès et al., 2009; Ganguli & Simoncelli, 2014; Wei & Stocker, 2016). Given the FIM, we can define the *sensitivity* of the

representation of stimulus $\mathbf{s}$ to a distortion $\boldsymbol{\epsilon}$ as:

$$d(\mathbf{s}; \boldsymbol{\epsilon}) := \sqrt{\boldsymbol{\epsilon}^\top \boldsymbol{I}(\mathbf{s}) \boldsymbol{\epsilon}}, \tag{1}$$

which quantifies how well an ideal observer could detect small perturbations of the base stimulus in the direction $\boldsymbol{\epsilon}$.

As a tractable example, suppose that the conditional response $\mathbf{r}$ is Gaussian with stimulus-dependent mean $\boldsymbol{f}(\mathbf{s})$ and constant covariance $\boldsymbol{\Sigma}$; that is, $p(\mathbf{r}|\mathbf{s}) \sim \mathcal{N}(\boldsymbol{f}(\mathbf{s}), \boldsymbol{\Sigma})$. Then the FIM at $\mathbf{s}$ is

$$\boldsymbol{I}(\mathbf{s}) = \boldsymbol{J}_f(\mathbf{s})^\top \boldsymbol{\Sigma}^{-1} \boldsymbol{J}_f(\mathbf{s}),$$

where $\boldsymbol{J}_f(\mathbf{s})$ is the Jacobian of $\boldsymbol{f}(\cdot)$ at $\mathbf{s}$. From this expression, we see that the sensitivities of a representation to a distortion $\boldsymbol{\epsilon}$ depend on how the mean response is changing in the direction $\boldsymbol{\epsilon}$ relative to the noise covariance. This example is relevant to our experimental results, where the function $\boldsymbol{f}(\mathbf{s})$ denotes the output of a deterministic model evaluated at a stimulus $\mathbf{s}$.

## 2.2 EIGEN-DISTORTIONS OF AN IMAGE REPRESENTATION

Given a model image representation, Berardino et al. (2017) proposed the use of extremal eigenvectors of the model FIM (termed "eigen-distortions", Fig. 1B, top panel) as model predictions of the most- and least-noticeable image distortions. For a set of early vision models and deep neural networks whose parameters were optimized to match a database of human image quality ratings (Ponomarenko et al., 2009), they computed the eigen-distortions of each model. Despite the fact that these models all fit the image quality data equally well, their eigen-distortions were quite different, and human perceptual judgements of the severity of these eigen-distortions varied substantially. The eigen-distortions of each image representation correspond to its most distinct distortion predictions, which can then be tested with human perceptual experiments. However, if the eigen-distortions of two models are similar, they will not be useful in distinguishing the models, since this method is insensitive to differences in the non-extremal eigenvectors.

## 2.3 GENERALIZED EIGEN-DISTORTIONS FOR COMPARING TWO IMAGE REPRESENTATIONS

Zhou et al. (2023) proposed comparing two image representations $A$ and $B$ along distortions in which their local sensitivities maximally differ, which is conceptually similar to previous methods that construct stimuli that maximize disagreement between models (Wang & Simoncelli, 2008; Golan et al., 2020). Specifically, they chose distortions to extremize the generalized Rayleigh quotient:

$$\boldsymbol{\epsilon}_1(\mathbf{s}) = \arg\max_{\mathbf{u}} \frac{\mathbf{u}^\top \boldsymbol{I}_A(\mathbf{s})\mathbf{u}}{\mathbf{u}^\top \boldsymbol{I}_B(\mathbf{s})\mathbf{u}}, \qquad\qquad \boldsymbol{\epsilon}_2(\mathbf{s}) = \arg\min_{\mathbf{v}} \frac{\mathbf{v}^\top \boldsymbol{I}_A(\mathbf{s})\mathbf{v}}{\mathbf{v}^\top \boldsymbol{I}_B(\mathbf{s})\mathbf{v}}. \tag{2}$$

Since these distortions correspond to the extremal eigenvectors of the generalized eigenvalue problem $\boldsymbol{I}_A(\mathbf{s})\boldsymbol{\epsilon} = \lambda\boldsymbol{I}_B(\mathbf{s})\boldsymbol{\epsilon}$, we refer to them as "generalized eigen-distortions" (Fig. 1B, middle panel). However, this method is limited to comparisons of pairs of models, or of a single model against the average of other models.

## 3 PRINCIPAL DISTORTIONS OF IMAGE REPRESENTATIONS

We propose a natural extension of generalized eigen-distortions that allows for comparisons among more than two image representations. We show that the generalized eigenvalue problem suggests a metric on the local geometry of image representations, which can be used to optimally choose image distortions that distinguish more than two models at a base stimulus.

## 3.1 A METRIC ON THE LOCAL GEOMETRY OF IMAGE REPRESENTATIONS

We can re-express the generalized eigen-distortions defined in Equation 2 as the solutions of a single optimization problem:

$$\{\boldsymbol{\epsilon}_1, \boldsymbol{\epsilon}_2\} = \arg\max_{\mathbf{u},\mathbf{v}} \left| \log \frac{d_A(\mathbf{u})}{d_B(\mathbf{u})} - \log \frac{d_A(\mathbf{v})}{d_B(\mathbf{v})} \right|.$$

where we've used the definition of the sensitivity in equation 1 and simplified notation by omitting the dependence of sensitivities and distortions on the stimulus, $\mathbf{s}$. We can regroup the numerators and denominators of the log quotients by model rather than by distortion, to re-express the optimization problem as follows:

$$\{\boldsymbol{\epsilon}_1, \boldsymbol{\epsilon}_2\} = \arg \max_{\mathbf{u}, \mathbf{v}} m_{\mathbf{u},\mathbf{v}}(\boldsymbol{I}_A, \boldsymbol{I}_B),$$

where

$$m_{\mathbf{u},\mathbf{v}}(\boldsymbol{I}_A, \boldsymbol{I}_B) := \left| \log \frac{d_A(\mathbf{u})}{d_A(\mathbf{v})} - \log \frac{d_B(\mathbf{u})}{d_B(\mathbf{v})} \right|. \tag{3}$$

For any pair of distortions $\mathbf{u}, \mathbf{v}$, the function $m_{\mathbf{u},\mathbf{v}}(\cdot, \cdot)$ defines a (pseudo-)metric.[1] Specifically, it is non-negative, symmetric, obeys the triangle inequality, and is zero when $\boldsymbol{I}_A = \boldsymbol{I}_B$. This metric has several appealing properties:

- Invariance to scaling of the FIMs by positive constants $c_A, c_B > 0$:
$$m_{\mathbf{u},\mathbf{v}}(\boldsymbol{I}_A, \boldsymbol{I}_B) = m_{\mathbf{u},\mathbf{v}}(c_A \boldsymbol{I}_A, c_B \boldsymbol{I}_B).$$
This is a desirable property since we are interested in identifying relevant image distortions that depend on the shape of the FIMs, independent of scaling factors.
- Invariance to permutation of the distortions $\mathbf{u}$ and $\mathbf{v}$:
$$m_{\mathbf{u},\mathbf{v}}(\boldsymbol{I}_A, \boldsymbol{I}_B) = m_{\mathbf{u},\mathbf{v}}(\boldsymbol{I}_A, \boldsymbol{I}_B).$$
- When $\boldsymbol{\epsilon}_1$ and $\boldsymbol{\epsilon}_2$ are the generalized eigen-distortions of $\boldsymbol{I}_A$ and $\boldsymbol{I}_B$, $m_{\boldsymbol{\epsilon}_1, \boldsymbol{\epsilon}_2}(\boldsymbol{I}_A, \boldsymbol{I}_B)$ is an approximation of the Fisher-Rao distance between mean-zero Gaussian distributions with covariances $\boldsymbol{I}_A$ and $\boldsymbol{I}_B$ (up to scaling factors, see Appx. A). This interpretation suggests a principled extension of the metric to more than two distortions.
- The metric compares stochastic representations back in stimulus space via their FIMs, which avoids having to align the stochastic representations (Duong et al., 2023b).

### 3.2 PRINCIPAL DISTORTIONS FOR COMPARING MULTIPLE IMAGE REPRESENTATIONS

To optimize a pair of image distortions for distinguishing $N > 2$ representations, $A_1, \ldots, A_N$, we choose $\boldsymbol{\epsilon}_1, \boldsymbol{\epsilon}_2$ to maximize the sum of the squares of all pairwise distances between the FIMs under the metric defined in Equation 3:

$$\{\boldsymbol{\epsilon}_1, \boldsymbol{\epsilon}_2\} = \arg \max_{\mathbf{u}, \mathbf{v}} \sum_{n=1}^{N} \sum_{m=1}^{N} m_{\mathbf{u},\mathbf{v}}^2 (\boldsymbol{I}_{A_n}, \boldsymbol{I}_{A_m}). \tag{4}$$

This is equivalent to maximizing the *variance* of the image representations' log sensitivity ratios:

$$\{\boldsymbol{\epsilon}_1, \boldsymbol{\epsilon}_2\} = \arg \max_{\mathbf{u}, \mathbf{v}} \sum_{n=1}^{N} \left| \log \frac{d_{A_n}(\mathbf{u})}{d_{A_n}(\mathbf{v})} - \frac{1}{N} \sum_{m=1}^{N} \log \frac{d_{A_m}(\mathbf{u})}{d_{A_m}(\mathbf{v})} \right|^2$$

We refer to $\{\boldsymbol{\epsilon}_1, \boldsymbol{\epsilon}_2\}$ as the "principal distortions" of the models, analogous to principal component analysis (Fig. 1B). For a gradient-based optimization algorithm, see Appx. B.

There are several other natural extensions of generalized-eigendistortions when considering $N > 2$ models. For example, for any $p \geq 1$, one can choose distortions $\boldsymbol{\epsilon}_1, \boldsymbol{\epsilon}_2$ that maximize the sum of the $p^{\text{th}}$ power of all pairwise distances. Here we focus on the case of maximizing the variance ($p = 2$) and leave an exploration of other moments to future work.

## 4 EXPERIMENTAL RESULTS

As a demonstration of our method, we generated principal distortions for computational models previously proposed to capture aspects of the human visual system. All models were implemented in PyTorch (Ansel et al., 2024) and simulations were performed on NVIDIA GPUs (RTX A6000 and A100 models). As the models are deterministic, we calculate the FIM by assuming the network output is corrupted by additive Gaussian noise, as in (Berardino et al., 2017). In this case, $\boldsymbol{I}(\mathbf{s}) = \boldsymbol{J}_f(\mathbf{s})^\top \boldsymbol{J}_f(\mathbf{s})$, where $\boldsymbol{J}_f(\mathbf{s})$ is the Jacobian of the model $\boldsymbol{f}(\cdot)$ at input $\mathbf{s}$ and the induced geometry on stimulus space is the pullback via $\boldsymbol{f}(\cdot)$ of the Euclidean geometry on representation space.

---

[1]Technically, $m_{\mathbf{u},\mathbf{v}}(\cdot, \cdot)$ is a *pseudometric* because $m_{\mathbf{u},\mathbf{v}}(\boldsymbol{I}_A, \boldsymbol{I}_B) = 0$ does not imply that $\boldsymbol{I}_A = \boldsymbol{I}_B$.

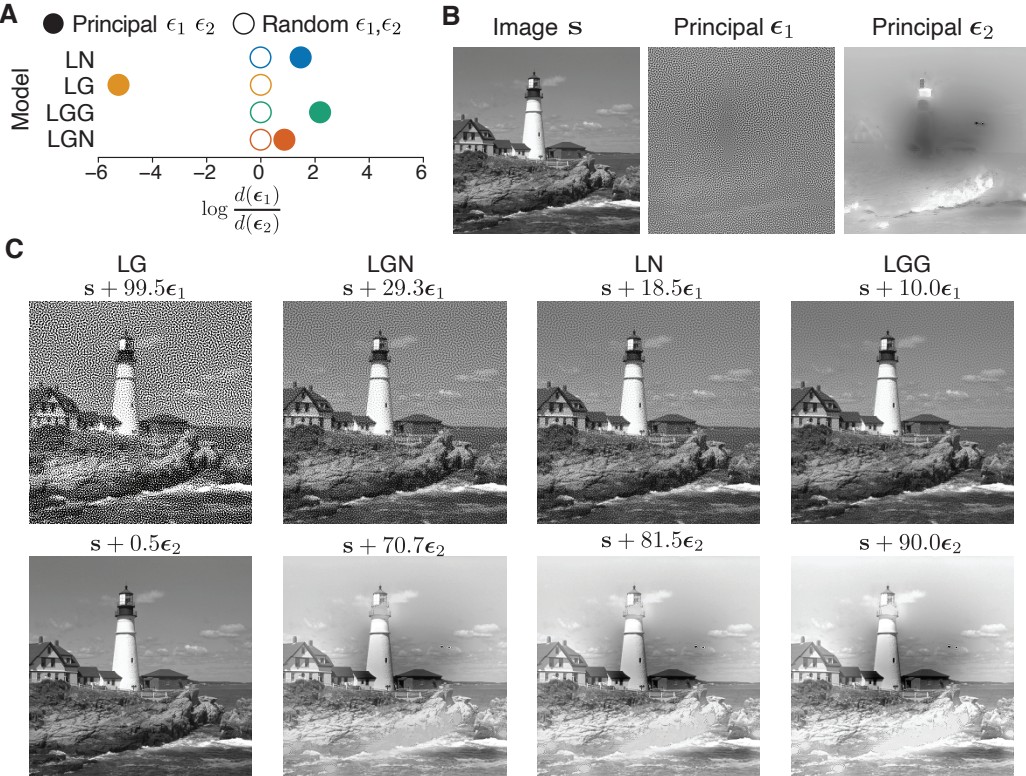

Figure 2: Principal distortions of four early visual models. **A)** Log sensitivity ratios of the two principal distortions and two random distortions for each model (Berardino et al., 2017). Models are nested (LN is the most basic, LGN is the full model). Principal distortions (filled circles) give rise to diverse log ratios, while random distortions (hollow circles) do not. **B)** Natural image $\mathbf{s}$ and corresponding optimized principal distortions $\{\boldsymbol{\epsilon}_1, \boldsymbol{\epsilon}_2\}$. **C)** Natural image corrupted by principal distortions, with each pair scaled so as to be equally detectable by one model (as indicated above). Models are ordered by the log ratio of their sensitivities (panel A). If a model's thresholds are proportional to human thresholds, the corresponding pair of scaled distortions should be equally visible in the top and bottom images. Note: Images are best viewed at high resolution.

## 4.1 EARLY VISUAL MODELS

We generated principal distortions for a nested family of models designed to capture the response properties of early stages (specifically, the lateral geneiculate nucleus, LGN) of the primate visual system (Fig. 2). The full model (LGN) contains two parallel cascades representing ON and OFF center-surround filter channels, rectification, and both luminance and contrast gain control nonlinearities. The other models are reduced versions of this model. LGG removes the OFF channel, LG additionally removes the contrast gain control, and LN removes both gain control operations. The filter sizes, amplitudes, and normalization values of each model were previously fit separately to predict a dataset of human distortion ratings (Berardino et al., 2017, see details in Appx. C).

As these models were explicitly trained to predict human distortion thresholds, we provide a qualitative comparison of each model's sensitivities to human distortion sensitivity (Fig. 2C). For each model, we adjusted the relative scaling of each principal distortion until the model was equally sensitive to the scaled distortions while constraining the sum of the Euclidean norms of the two distortions to be a fixed value of 100; that is, we chose positive scalars $k_1, k_2$ such that the sensitivities were equal, $d(k_1\boldsymbol{\epsilon}_1) = d(k_2\boldsymbol{\epsilon}_2)$, with $k_1 + k_2 = 100$. If a model's thresholds are comparable to human thresholds, then the pair of images should also appear equally distorted to a human observer. Visual inspection of these images reveals that both distortions are visible when rescaled for the LGN model and the LN model, suggesting that these models are closest to human distortion thresholds. For LG, the scaled $\boldsymbol{\epsilon}_2$ distortion is not visible, while the scaled $\boldsymbol{\epsilon}_1$ distortion is immediately apparent, suggesting a strong mismatch with human observer thresholds. The same is true of the LGG model,

with the roles of the two distortions swapped. These qualitative observations are consistent with the results of (Berardino et al., 2017), in which experiments on eigen-distortions suggested that the LGN model was the best of these models in terms of consistency with human distortion sensitivity. Future work with analogous human perceptual experiments could directly quantify the visibility of the principal distortions arising from our analysis (see Supp. Fig. SI.1 for an illustration).

An advantage of our framework is that it can dramatically reduce the number of distortions needed to differentiate a set of models. For instance, to judge how well these models of the early visual system capture human perceptual discrimination thresholds, Berardino et al. (2017) computed the extremal eigen-distortions for each of the four models, and then measured human perceptual discrimination thresholds to all eight distortions. In general, their method requires assessing visibility of $2N$ distortions, for $N$ models. Zhou et al. (2023) considered a pair of generalized eigen-distortions for each pair of models, for a total of $N(N + 1)$ distortions. In contrast, our method always selects the two distortions that maximize the variance across the models, independent of $N$. Human sensitivities to these distortions can then be estimated in perceptual discrimination experiments to judge which model(s) are closest in terms of the metric we defined Equation 3. The models whose sensitivities are far from human sensitivities can be discarded and this procedure can be repeated to best differentiate the remaining models, and so on. If one could reduce the number of models by, say, a factor of two on each iteration of this process, the total number of stimuli to be assessed scales as $2 \log_2(N)$. This dramatic improvement in efficiency could enable the comparison of significantly larger sets of models than feasible with previous methods.

## 4.2 DEEP NEURAL NETWORKS

Deep Neural Networks (DNNs), originally developed for object recognition, have also been examined as models of the primate visual system (Yamins & DiCarlo, 2016; Schrimpf et al., 2018; Lindsay, 2021). A plethora of models, varying in architecture and training techniques, have been proposed, but many of these models perform quite similarly on behavioral tasks or neural benchmarks (Schrimpf et al., 2018; Tuckute et al., 2023; Conwell et al., 2024). This situation offers a well-aligned opportunity for use of our principal distortion method. Here, we investigate previously trained sets of models (no additional training is required to apply our method) and generate principal distortions to differentiate the image representations.

We first investigated a set of layers from two architectures trained on the ImageNet object classification task—AlexNet (Krizhevsky et al., 2012) and ResNet50 (He et al., 2016)—and generated the principal distortions that maximally separate these models (Fig. 3, see Sec. D of the supplement for layer choices and model details). Although these architectures are not currently state-of-the-art at image recognition or neural prediction, they have been widely used and trained with various optimization strategies, making them well suited for controlled experiments. (See Supp. Fig. SI.2 for a similar test of the method on the modern architectures EfficientNet and ViT.)

Notably, the hierarchical structure of the models is reflected in the log ratios of the sensitivities, where early layers of the models are closer together in the metric space (Fig. 3D), and late layers of AlexNet are always more sensitive to $\epsilon_1$ when late layers of ResNet50 are more sensitive to $\epsilon_2$. There is additional structure revealed by these principal distortions—AlexNet is more sensitive to the principal distortion that generally appears concentrated on regions of the image that have more variability, i.e., the "stuff" in the image (distortion $\epsilon_1$ in Fig. 3B), while ResNet50 is more sensitive to distortions that occur in the relatively constant regions of the image (distortion $\epsilon_2$ in Fig. 3B). The separability of the models and the sensitivity of the networks to the two distortion types was remarkably consistent across a set of 100 base images chosen from the ImageNet dataset (Fig. 3E, see Supp. Fig. SI.3 for additional examples). These distortions differentially affected the classification decisions of the two architectures (Supp. Fig. SI.4). The distortions also replicated in a set of six models (3 randomly initialized AlexNet models and 3 randomly initialized ResNet50 models, each trained on ImageNet-1k, see Supp. Fig. SI.5). This distinction was also found in images designed explicitly to test for possible differences between models due to edge artifacts or contrast sensitivity (Supp. Fig. SI.6). As far as we know, this qualitative difference in sensitivities of the architectures to distortions in different portions of the image has not been documented, demonstrating that our method can reveal interpretable differences in the local sensitivities of complex computational models.

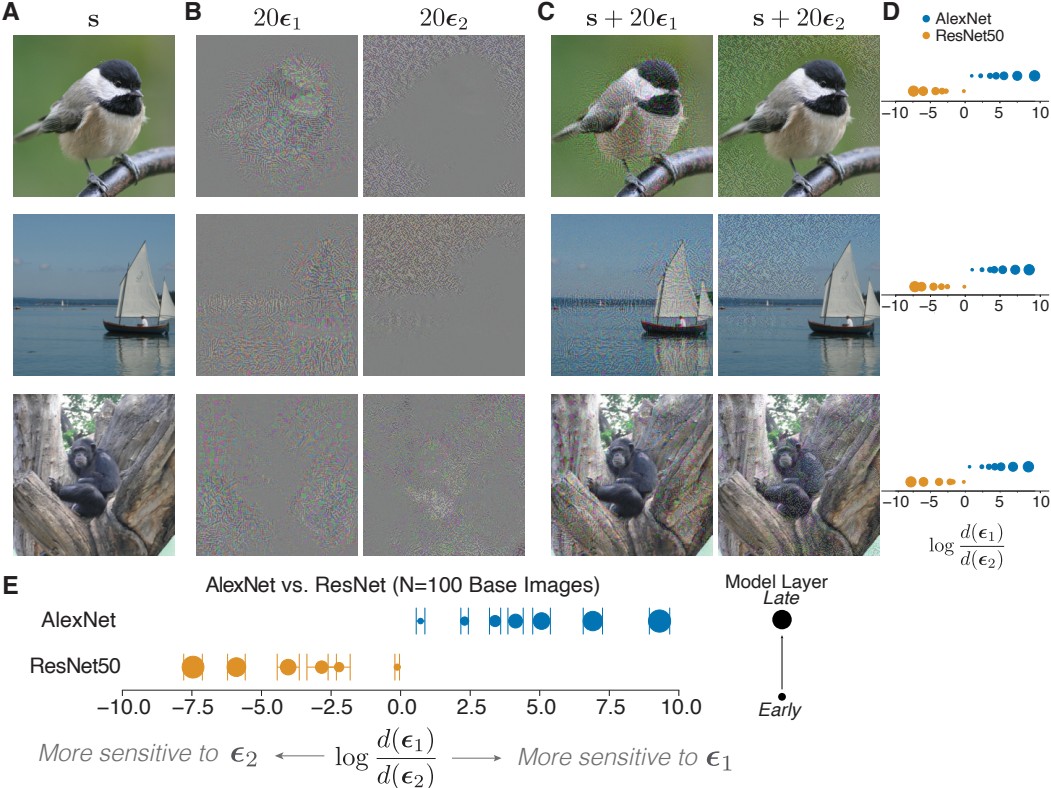

Figure 3: AlexNet versus ResNet50. **A)** Example base images. **B)** Optimized principal distortions (scaled by a factor of 20 for visibility, and using the convention $\|\epsilon\| = 1$ here and in other figures). **C)** Base image plus principal distortions. **D)** Log sensitivity ratios of principal distortions when comparing image representations at multiple layers of AlexNet and ResNet50. Assignment of $\epsilon_1$ and $\epsilon_2$ was chosen so that the final tested layer of AlexNet has a positive log ratio. **E)** Log sensitivity ratios averaged across 100 base images (error bars are standard deviation). The principal distortions organize the networks by architecture—the log sensitivity ratios of AlexNet and ResNet50 are separated and early layers have smaller log ratios than late layers. AlexNet is more sensitive to distortion $\epsilon_1$, which is concentrated on higher contrast or textured parts of the image (often the foreground object). ResNet50 is more sensitive to distortion $\epsilon_2$ which concentrates power on relatively smooth parts of the image, such as regions of constant intensity/color.

**Networks trained to reduce texture bias** The architectural difference observed between ImageNet-trained AlexNet and ResNet50 suggested that the texture of the image may be driving some of the differences in local geometry. Previous work demonstrated that standard DNNs exhibit strong "texture bias" (Geirhos et al., 2019) and proposed models that explicitly reduce the texture bias by training on Stylized ImageNet (SIN), a set of images that retains the content of each ImageNet image but overlays details matched to particular texture. Training on these images can reduce the model's reliance on texture for classification. If this training set strongly affected the local geometry of the networks, we might expect that principal distortions generated for a mixture of architectures and training sets would be driven by this training set distinction. We find evidence that this is not the case. We generated principal distortions for 100 base images using a set of layers from two architectures (ResNet50 and AlexNet) and two different training datasets (ImageNet and SIN, Fig. 4A). The principal distortions appeared qualitatively similar to those generated when we investigated only the standard networks (Fig. 4B, more examples in Supp. Fig. SI.7). Both AlexNet architectures, regardless of the training type, were more sensitive to the perturbations of higher variability parts of the image, while both ResNet50 models were more sensitive to the distortions that were mainly targeting constant areas of the image. To quantify this observation and dissociate the role of foreground and background information from the high- and low-frequency content in the image, we ran an experiment using the ImageNet-9 dataset with foreground and background masks, where we generated principal distortions after applying a low-pass filter to either the foreground

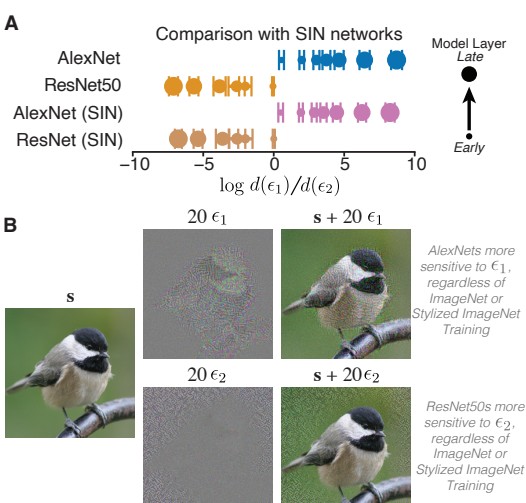

Figure 4: Comparison of AlexNet and ResNet50 variants trained to increase "Shape Bias" of networks. **A)** Log sensitivity ratios of principal distortions for networks trained on standard ImageNet and Stylized ImageNet (SIN). Average log sensitivity is computed across 100 base images (error bars are standard deviation) and the choice of $\epsilon_1$ is set such that $d(\epsilon_1) \geq 0$ for the last tested layer of the standard AlexNet. **B)** Example base image and the optimized principal distortions. Similar to the analysis of the ImageNet-trained ResNet50 and AlexNet (Fig. 3), $\epsilon_1$ is concentrated on parts of the image with higher spatial frequencies while $\epsilon_2$ is concentrated on parts of the image with relatively solid patches. Both AlexNet architectures are more sensitive to $\epsilon_1$ while both ResNet50 architectures are more sensitive to $\epsilon_2$, suggesting that the differences in local sensitivities of these networks depend more on differences in architecture than training procedure.

or the background of the image (Supp. Fig. SI.8). The principal distortion that AlexNet is more sensitive to consistently concentrated on portions of the image with higher spatial frequencies (i.e., parts of the image that are not smoothed), while ResNet50 is concentrated on parts of the image with only low-frequency information (the blurred parts of the image, regardless of whether this is in the foreground or background).

**Networks trained to reduce adversarial vulnerability**   Another well-known example of the use of training set modifications to achieve robustness in object recognition is that of adversarial training (AT). Adversarial examples are generated at each step of model training and these stimuli are used to update the model weights, with the "true" category label used for the update. Previous work has found that adversarially-trained networks are more aligned with those of biological systems (Madry, 2017; Feather et al., 2023; Gaziv et al., 2023). As adversarial examples are constrained to be very small perturbations, it seems plausible that the local geometry for AT models would differ from their standard counterparts. Indeed, we see that the principal distortions generated from the set of models that included standard and AT trained AlexNet and ResNet50 models reliably separate the model classes by training type, rather than architecture (Fig. 5A). Most layers of the adversarially trained models are more sensitive to relatively smooth changes of patches of color in the image, or to shading around edges, while most layers of the standard models are more sensitive to what appears as unstructured noise (Fig. 5B, additional examples in Supp. Fig. SI.9). This is in stark contrast to the SIN-trained networks of the previous section, for which principal distortions reliably separated the models by architecture. These examples demonstrate that our method can be used to separate collections of similar models, and points to its utility in probing complex high-level representations.

## 5   DISCUSSION

We have introduced a metric on image representations that quantifies differences in local geometry, and used it to synthesize "principal distortions" that maximize the variance of the metric over a set of models. When applied to hand-engineered models of the early visual system and to DNNs, our approach produced novel distortions for distinguishing the corresponding models. In particular, with the DNN analysis, we revealed that there are qualitative differences in local geometries of ResNet50 and AlexNet architectures, that adversarial training dramatically changes the local geometry, but stylized ImageNet training does not. Although our qualitative examples in this targeted set of neural networks do not fully elucidate the recent observations that many different models are equally good at capturing brain representations, our method provides a direct approach to begin to tease apart the interplay between local geometry and global structure.

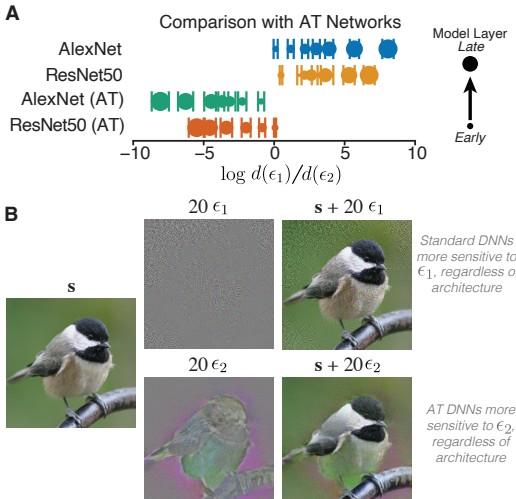

Figure 5: Comparison of AlexNet and ResNet50 variants trained to reduce adversarial vulnerability. **A)** Log sensitivity ratios of principal distortions for different layers of standard-trained and adversarially-trained (AT) models. Average log sensitivity is computed across 100 base images (error bars are standard deviation) and the choice of $\epsilon_1$ is set such that $d(\epsilon_1) \geq 0$ for the last tested layer of the standard AlexNet. **B)** Example base image and the generated principal distortions. Distortion $\epsilon_1$ appears as less structured noise, and both AlexNet and ResNet50 standard networks are more sensitive to this perturbation, while the AT DNNs are more sensitive to $\epsilon_2$ which focuses color changes around the content of the image, suggesting that the differences in local sensitivities of these networks depend more on differences in training procedure than architecture.

There are several natural methodological extensions. The metric is closely related to the Fisher-Rao metric between mean zero Gaussians and this relation suggests a natural extension to synthesizing more than two distortions (Appx. A), analogous to using additional principal components to capture more variance within a set of high-dimensional vectors. Additionally, while we focus on computing principal distortions that differentiate a finite set of models, there is a natural extension to computing the principal distortions that differentiate continuous families of models with a prior distribution over models.

There are several limitations in our formulation of principal distortions. First, our framework is based on local differential analyses of a model at a base image, so the sensitivity estimates we obtain via these analyses can only be guaranteed to hold in an infinitesimally small neighborhood of the base image. If a model is highly nonlinear in the vicinity of a base image, then the local linear approximation may not accurately reflect model sensitivities. Second, to compute the FIM of a deterministic model, we assume additive Gaussian response noise, which is not generally representative of neural responses in the brain. Poisson variability could be used to better capture neural responses. Another approach is to fit the model noise structure to measurements of a neural system; see (Ding et al., 2023; Nejatbakhsh et al., 2023) for work along these lines.

Principal distortions provide an efficient method for comparing computational models with human observers, for whom experimental time for acquiring responses to stimuli is generally severely limited. Although we only presented qualitative comparisons and examples related to human perception in this paper, the optimized distortions are a parsimonious choice of stimuli that can be readily incorporated into psychophysics experiments. For example, the distortions generated from the early visual models could be used for a perceptual discrimination experiment with human observers similar to those performed by Berardino et al. (2017), to compare the human log-ratio sensitivity for the optimized distortions to that predicted by the models.

The results with DNNs reveal some intriguing properties: some models have stronger sensitivity biases in the local geometry for perturbing higher frequency regions of the space, while others have more sensitivity to perturbations in relatively low frequency regions. This also suggests an interesting question for future distortion detection: in what contexts are humans more sensitive to perturbations of the "stuff" of the image compared to perturbations in empty parts of the image? And how do these observed differences relate to previous work investigating how human and neural networks rely on spatial frequencies for classification (Subramanian et al., 2023)? Finally, beyond direct comparisons with human observers using psychophysics experiments, our method may be useful in the domain of neural network interpretability, where it may be useful to have direct comparisons of the local distortions that will maximally differentiate sets of models. Principal distortions are complementary to many currently used model interpretability tools (e.g., saliency maps, attention weight visualization) since they are explicitly designed to generate distortions that differentiate a set of models, as opposed to interpreting one model at a time.

## ACKNOWLEDGMENTS

The Flatiron Institute is a division of the Simons Foundation. The computations reported in this paper were performed using resources made available by the Flatiron Institute. We thank David Brainard, Thomas Yerxa, and the Laboratory for Computational Vision at NYU and the Flatiron Instiute for their feedback.

## REPRODUCIBILITY STATEMENT

The code used to generate principal distortions, details of loading the models, and example principal distortion generation code is available at

https://github.com/LabForComputationalVision/principal-distortions

We aim for the distortion generation to be easy for others to use to probe new models. The mathematical derivation of the algorithm is provided in Appx. B. We have included details in Appx. C with the parameters of the early visual models, and Appx. D details where checkpoints were obtained for the tested deep neural networks.

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

## A   RELATION TO THE FISHER-RAO METRIC

The Fisher-Rao distance between two mean zero $K$-dimensional Gaussian distributions with positive definite covariance matrices $\boldsymbol{A}$ and $\boldsymbol{B}$ is defined as:

$$\delta^2(\boldsymbol{A}, \boldsymbol{B}) := \| \log(\boldsymbol{B}^{-1/2} \boldsymbol{A} \boldsymbol{B}^{-1/2})\|_F^2 = \sum_{i=1}^{K} (\log \lambda_i)^2,$$

where $\{\lambda_i\}$ denote the eigenvalues of the generalized eigenvalue problem $\boldsymbol{A}\boldsymbol{v} = \lambda \boldsymbol{B}\boldsymbol{v}$ (Pinele et al., 2020). We'd like a metric that's invariant to arbitrary scalings $\boldsymbol{A} \mapsto c_A \boldsymbol{A}$ or $\boldsymbol{B} \mapsto c_B \boldsymbol{B}$ for $c_A, c_B > 0$, which suggests the following definition:

$$\gamma^2(\boldsymbol{A}, \boldsymbol{B}) = \min_{c_A, c_B > 0} \delta^2(c_A \boldsymbol{A}, c_B \boldsymbol{B}) = \min_{c \in \mathbb{R}} \sum_{i=1}^{K} (c + \log \lambda_i)^2 = K \mathrm{Var}(\{\log \lambda_i\}), \tag{5}$$

where the final equality uses the fact that the optimal $c$ is the mean of $\{-\log \lambda_i\}$.

Using the facts that

$$K \mathrm{Var}(\{\log \lambda_i\}) = \frac{1}{2K} \sum_{i=1}^{K} \sum_{j=1}^{K} (\log \lambda_i - \log \lambda_j)^2,$$

and $(\log \lambda_i - \log \lambda_j)^2 \leq (\log \lambda_1 - \log \lambda_K)^2$ for all $i, j$, we have

$$\frac{1}{K} (\log \lambda_1 - \log \lambda_K)^2 \leq \gamma^2(\boldsymbol{A}, \boldsymbol{B}) \leq \frac{K-1}{2} (\log \lambda_1 - \log \lambda_K)^2,$$

with equality holding when $K = 2$. When $\boldsymbol{A} = \boldsymbol{I}_A$ and $\boldsymbol{B} = \boldsymbol{I}_B$, then $d_A(\boldsymbol{\epsilon}) = \sqrt{\boldsymbol{\epsilon}^\top \boldsymbol{A} \boldsymbol{\epsilon}}$ and $d_B(\boldsymbol{\epsilon}) = \sqrt{\boldsymbol{\epsilon}^\top \boldsymbol{B} \boldsymbol{\epsilon}}$. If $\boldsymbol{\epsilon}_1$ and $\boldsymbol{\epsilon}_K$ denote the extremal generalized eigenvectors associated with $\lambda_1$ and $\lambda_K$, respectively, then

$$\log \lambda_1 = 2 \log \frac{d_A(\boldsymbol{\epsilon}_1)}{d_B(\boldsymbol{\epsilon}_1)}, \qquad\qquad \log \lambda_K = 2 \log \frac{d_A(\boldsymbol{\epsilon}_K)}{d_B(\boldsymbol{\epsilon}_K)}.$$

Therefore,

$$\frac{2}{\sqrt{K}} \left| \log \frac{d_A(\boldsymbol{\epsilon}_1)}{d_B(\boldsymbol{\epsilon}_1)} - \log \frac{d_A(\boldsymbol{\epsilon}_K)}{d_B(\boldsymbol{\epsilon}_K)} \right| \leq \gamma(\boldsymbol{A}, \boldsymbol{B}) \leq \sqrt{2(K-1)} \left| \log \frac{d_A(\boldsymbol{\epsilon}_1)}{d_B(\boldsymbol{\epsilon}_1)} - \log \frac{d_A(\boldsymbol{\epsilon}_K)}{d_B(\boldsymbol{\epsilon}_K)} \right|,$$

and so

$$\frac{2}{\sqrt{K}} m_{\boldsymbol{\epsilon}_1, \boldsymbol{\epsilon}_K}(\boldsymbol{I}_A, \boldsymbol{I}_B) \leq \gamma(\boldsymbol{A}, \boldsymbol{B}) \leq \sqrt{2(K-1)} m_{\boldsymbol{\epsilon}_1, \boldsymbol{\epsilon}_K}(\boldsymbol{I}_A, \boldsymbol{I}_B). \tag{6}$$

**Extension to more than two distortions**   Equation 5 suggests a natural extension for defining a metric between positive definite matrices using $J > 2$ distortions. Specifically, it suggests choosing the distortions to maximize the variance across the log ratios of the sensitivities. To this end, we can define the squared distance between the local geometries to be the variance (across *distortions*) of the log ratio of the the sensitivities:

$$m_{\mathbf{u}_1, \dots, \mathbf{u}_J}^2(\boldsymbol{A}, \boldsymbol{B}) = M \mathrm{Var} \left( \left\{ \log \frac{d_A(\mathbf{u}_j)}{d_B(\mathbf{u}_j)} \right\} \right).$$

A set of $J$ principal distortions can then be chosen to maximize the variance across *models* under this metric.

## B   COMPUTING THE TOP TWO OPTIMAL DISTORTIONS

Suppose we have $N$ models with sensitivities $\{d_n(\boldsymbol{\epsilon})\}$. The optimal distortions $\{\boldsymbol{\epsilon}_1, \boldsymbol{\epsilon}_2\}$ are solutions to the optimization problem

$$\underset{\mathbf{u}, \mathbf{v}}{\arg\max}\, L(\mathbf{u}, \mathbf{v}), \qquad L(\mathbf{u}, \mathbf{v}) := \sum_{n=1}^{N} \left\{ \log \frac{d_n(\mathbf{u})}{d_n(\mathbf{v})} - \frac{1}{N} \sum_{m=1}^{N} \log \frac{d_m(\mathbf{u})}{d_m(\mathbf{v})} \right\}^2.$$

Differentiating $L$ with respect to $\mathbf{u}$ yields

$$\nabla_{\mathbf{u}} L(\mathbf{u}, \mathbf{v}) = 2 \sum_{n=1}^{N} \left\{ \log \frac{d_n(\mathbf{u})}{d_n(\mathbf{v})} - \frac{1}{N} \sum_{m=1}^{N} \log \frac{d_m(\mathbf{u})}{d_m(\mathbf{v})} \right\} \left\{ \frac{\boldsymbol{I}_n(\mathbf{s})\mathbf{u}}{d_n^2(\mathbf{u})} - \frac{1}{N} \sum_{m=1}^{N} \frac{\boldsymbol{I}_m(\mathbf{s})\mathbf{u}}{d_m^2(\mathbf{u})} \right\}$$

$$= \sum_{n=1}^{N} \left\{ \log \frac{d_n^2(\mathbf{u})}{d_n^2(\mathbf{v})} - \frac{1}{N} \sum_{m=1}^{N} \log \frac{d_m^2(\mathbf{u})}{d_m^2(\mathbf{v})} \right\} \left\{ \frac{\boldsymbol{I}_n(\mathbf{s})\mathbf{u}}{d_n^2(\mathbf{u})} - \frac{1}{N} \sum_{m=1}^{N} \frac{\boldsymbol{I}_m(\mathbf{s})\mathbf{u}}{d_m^2(\mathbf{u})} \right\},$$

where we have used the fact that

$$\nabla_{\mathbf{u}} \log d(\mathbf{u}) = \frac{1}{2} \nabla_{\mathbf{u}} \log(\mathbf{u}^{\top} \boldsymbol{I} \mathbf{u}) = \frac{\boldsymbol{I}\mathbf{u}}{\mathbf{u}^{\top} \boldsymbol{I} \mathbf{u}} = \frac{\boldsymbol{I}\mathbf{u}}{d^2(\mathbf{u})}.$$

Similarly, differentiating $L$ with respect to $\mathbf{v}$ yields:

$$\nabla_{\mathbf{v}} L(\mathbf{u}, \mathbf{v}) = -\sum_{n=1}^{N} \left\{ \log \frac{d_n^2(\mathbf{u})}{d_n^2(\mathbf{v})} - \frac{1}{N} \sum_{m=1}^{N} \log \frac{d_m^2(\mathbf{u})}{d_m^2(\mathbf{v})} \right\} \left\{ \frac{\boldsymbol{I}_n(\mathbf{s})\mathbf{v}}{d_n^2(\mathbf{v})} - \frac{1}{N} \sum_{m=1}^{N} \frac{\boldsymbol{I}_m(\mathbf{s})\mathbf{v}}{d_m^2(\mathbf{v})} \right\}.$$

Combining, we have the following gradient-based optimization algorithm.

---

**Algorithm 1:** Computing the principal distortions via projected gradient descent

---

1: **Input:** Positive definite $D \times D$ matrices $\boldsymbol{I}_1, \ldots, \boldsymbol{I}_N$, learning rate $\eta > 0$, target distortion size $\alpha > 0$
2: **Initialize:** distortions $\mathbf{u}, \mathbf{v} \in \mathbb{R}^D$
3: **while** not converged **do**
4:     **for** $n = 1, \ldots, N$ **do**
5:         $\boldsymbol{v}_1(n) \leftarrow \boldsymbol{I}_n \mathbf{u}$
6:         $\boldsymbol{v}_2(n) \leftarrow \boldsymbol{I}_n \mathbf{v}$
7:         $d_1^2(n) \leftarrow \langle \mathbf{u}, \boldsymbol{v}_1(n) \rangle$
8:         $d_2^2(n) \leftarrow \langle \mathbf{v}, \boldsymbol{v}_2(n) \rangle$
9:         $\boldsymbol{u}_1(n) = \boldsymbol{v}_1(n)/d_1^2(n)$
10:        $\boldsymbol{u}_2(n) = \boldsymbol{v}_2(n)/d_2^2(n)$
11:        $r(n) \leftarrow \log d_1^2(n) - \log d_2^2(n)$
12:     **end for**
13:     $\bar{\boldsymbol{u}}_1 \leftarrow \text{mean}(\boldsymbol{u}_1(n))$
14:     $\bar{\boldsymbol{u}}_2 \leftarrow \text{mean}(\boldsymbol{u}_2(n))$
15:     $\bar{r} \leftarrow \text{mean}(r(n))$
16:     $\Delta\mathbf{u} \leftarrow \sum_{n=1}^{N} [r(n) - \bar{r}] [\boldsymbol{u}_1(n) - \bar{\boldsymbol{u}}_1]$
17:     $\Delta\mathbf{v} \leftarrow -\sum_{n=1}^{N} [r(n) - \bar{r}] [\boldsymbol{u}_2(n) - \bar{\boldsymbol{u}}_2]$
18:     $\mathbf{u} \leftarrow \mathbf{u} + \eta\Delta\mathbf{u}$
19:     $\mathbf{v} \leftarrow \mathbf{v} + \eta\Delta\mathbf{v}$
20:     $\mathbf{u} \leftarrow \alpha\mathbf{u}/\|\mathbf{u}\|$
21:     $\mathbf{v} \leftarrow \alpha\mathbf{v}/\|\mathbf{v}\|$
22: **end while**

---

## C    METHODS: EARLY VISUAL MODEL EXPERIMENTS

PyTorch implementations of the early visual models were obtained from (`https://github.com/plenoptic-org/plenoptic`, Duong et al., 2023a). The early visual models were adopted from Berardino et al. (2017), with parameters chosen to maximize the correlation between predicted perceptual distance and human ratings of perceived distortion, for a wide range of images and distortions provided in the TID-2008 database (Ponomarenko et al., 2009). Parameters are reported in Table 1. Note that although the models are nested in their construction and parameterization, the model parameters differ across tested models since they are optimized for each model independently.

### C.1   DATASET

Distortions were generated for images from the Kodak TID 2008 dataset (Ponomarenko et al., 2009).

| LN Model | |
|---|---|
| center-surround, center std | 0.5339 |
| center-surround, surround std | 6.148 |
| center-surround, amplitude ratio | 1.25 |
| **LG Model** | |
| luminance, scalar | 14.95 |
| luminance, std | 4.235 |
| center-surround, center std | 1.962 |
| center-surround, surround std | 4.235 |
| center-surround, amplitude ratio | 1.25 |
| **LGG Model** | |
| luminance, scalar | 2.94 |
| contrast, scalar | 34.03 |
| center-surround, center std | 0.7363 |
| center-surround, surround std | 48.37 |
| center-surround, amplitude ratio | 1.25 |
| luminance, std | 170.99 |
| contrast, std | 2.658 |
| **LGN Model** | (Two channels) |
| luminance, scalar | [3.2637, 14.3961] |
| contrast, scalar | [7.3405, 16.7423] |
| center-surround, center std | [1.237, 0.3233] |
| center-surround, surround std | [30.12, 2.184] |
| center-surround, amplitude ratio | 1.25 |
| luminance, std | [76.4, 2.184] |
| contrast, std | [7.49, 2.43] |

Table 1: Parameters for early visual models, obtained from (Berardino et al., 2017).

## D METHODS: DEEP NEURAL NETWORK EXPERIMENTS

With the exception of the experiment described in the next paragraph, we analyzed DNNs were obtained from the model loading code and checkpoints available at `https://github.com/jenellefeather/model_metamers_pytorch` which were used in (Feather et al., 2023), and allowed for easy loading and selection of the intermediate layer stages for many models that had previously been proposed as models of human visual perception. The checkpoints for the standard ResNet50 and AlexNet models were obtained from the public PyTorch checkpoints (Ansel et al., 2024). The Stylized Image Net AlexNet (`alexnet_trained_on_SIN`) and ResNet50 (`resnet50_trained_on_SIN`) were obtained from `https://github.com/rgeirhos/texture-vs-shape` associated with (Geirhos et al., 2019). The checkpoint for the ResNet50 $\ell_2(\epsilon_{\text{train}} = 3.0)$ adversarially trained model was obtained from (Engstrom et al., 2019), and the checkpoint for the Alexnet $\ell_2(\epsilon_{\text{train}} = 3.0)$ adversarially trained model was obtained from (Feather et al., 2023). For all experiments, we only included intermediate layers before the final classification stage in the principal distortion analysis, and the subset of layers followed those chosen in Feather et al. (2023). Specifically, for the AlexNet models we included layers `relu0`, `relu1`, `relu2`, `relu3`, `relu4`, `fc0_relu`, and `fc1_relu` in each set of anlayses. For the ResNet50 models we included `conv1_relu1`, `layer1`, `layer2`, `layer3`, `layer4`, and `avgpool` in each set of analyses.

To demonstrate that our approach holds for more modern architectures, we compared a ViT vs. and EfficientNet model. These models were obtained from the PyTorch Image Models (timm) repository (Wightman, 2019). We chose a version of EfficientNet and ViT that were trained on the size of Images in ImageNet-9 (224 × 224 images), and versions of each architecture trained with the same image dataset (ImageNet-1k). For the EfficientNet, we tested model EfficientNet-B0 (timm model `tf_efficientnet_b0`) and included layers `conv_stem`, `blocks.0`, `blocks.1`, `blocks.2`, `blocks.3`, `blocks.4`, `blocks.5`, `blocks.6`, and `global_pool` in the analysis. For the ViT, we tested model ViT (Base-Patch16-224) (timm model `vit_base_patch16_224.augreg_in1k`) and included layers `patch_embed`, `blocks.0`,

```
blocks.1, blocks.2, blocks.3, blocks.4, blocks.5, blocks.6, blocks.7,
blocks.8, blocks.9, blocks.10, blocks.11
```
and `head` in the analysis.

## D.1 DATASETS

For the deep neural network experiments we had two sets of images that were used to generate the principal distortions. For the main experiments, we use a subset of 100 ImageNet images where each image was chosen from a unique class (randomly chosen from the set of images at `https://github.com/EliSchwartz/imagenet-sample-images`).

For the experiments with background and foreground blur, we used the ImageNet-9 dataset with foreground/background masks (Xiao et al., 2021). We chose eight random images from each category of images, resulting in 72 total images. As this is a relatively small subset of images, we removed images that had blank backgrounds (by eye) or that seemed to have incorrect masks (as the masks were automatically generated in the original dataset and not validated with human labeling) and replaced them with new random images so that the backgrounds would always be affected by the smoothing procedure. This procedure was done before running the images through the principal distortion analysis. For the foreground and background blurred images, we blurred the image with a Gaussian filter with a standard deviation of 5 pixels.

## D.2 PRINCIPAL DISTORTION OPTIMIZATION

For each image and each comparison, we ran the gradient descent procedure for principal distortion optimization for 2500 iterations, using an exponentially decaying learning rate that started at 10.0 and decayed to 0.001 by the final step. The exception to this is for the experiment with ViT (Base-Patch16-224) vs. EfficientNet-B0, where we ran the optimization for only 500 iterations.

We used a target distortion size of $\alpha = 0.1$ and at each step of the optimization, we also scaled $\epsilon$ so that the image $s + 1000\epsilon$ would not be clipped when the RGB value was represented between 0–1; that is, we scaled $\epsilon$ so that $0 \leq s[i] + 1000\epsilon[i] \leq 1$ for each value $i$ in the image. This constraint could potentially bias the perturbations to be more spread out across the image (because all of the amplitude for the perturbation cannot be focused at a small set of pixels). However, removing this constraint did not lead to quantitatively different results (Supp. Fig. SI.10), but the inclusion allows for more viable comparisons with human perception (as the perturbation can be scaled while maintaining valid values in the image gamut). Finally, if one or more of the FIMs is degenerate (i.e., not full rank), this can potentially lead to issues in the optimization procedure (though this issue did not arise in the experiments presented here). To address this, one option is to regularize the FIM by adding a small constant times the identity matrix.

## D.3 SUPPLEMENTAL FIGURES

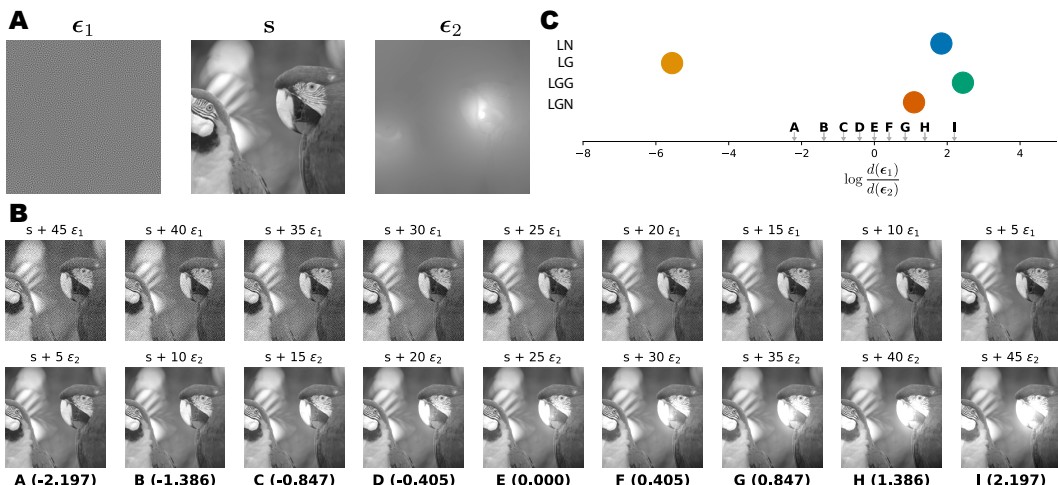

Figure SI.1: Illustration of method for comparing early visual models to human perception. **A)** An example base image (**s**) and the principal distortions ($\epsilon_1$, $\epsilon_2$) generated for the four early visual models (as in Fig. 2). **B)** Base image corrupted by the two principal distortions (top and bottom row), scaled by amplitudes that sum to 50 and correspond to different log sensitivity ratios (parenthesized values below the pairs, also indicated with grey arrows in panel **C**). A perceptual experiment can be designed to test which pair of corrupted images (indexed A through I) appear equally distorted to a human observer. This could be done directly, by showing various pairs of distorted images (i.e., $\mathbf{s} + k_1\epsilon_1$ and $\mathbf{s} + k_2\epsilon_2$ with varying amplitudes $k_1, k_2$) and asking an observer which of the two appears more distorted with respect to the original (**s**). The pair of images for which an observer cannot decide (i.e., gives random answers) corresponds to the observer's log sensitivity ratio. Alternatively, this could be assessed by measuring the observer's detection thresholds for each distortion independently, and then taking their log ratio. This estimated human log sensitivity ratio can then be compared to the models' log sensitivity ratios (colored dots, panel **C**), to assess which model is best aligned with human behavior. Images are best viewed at high resolution.

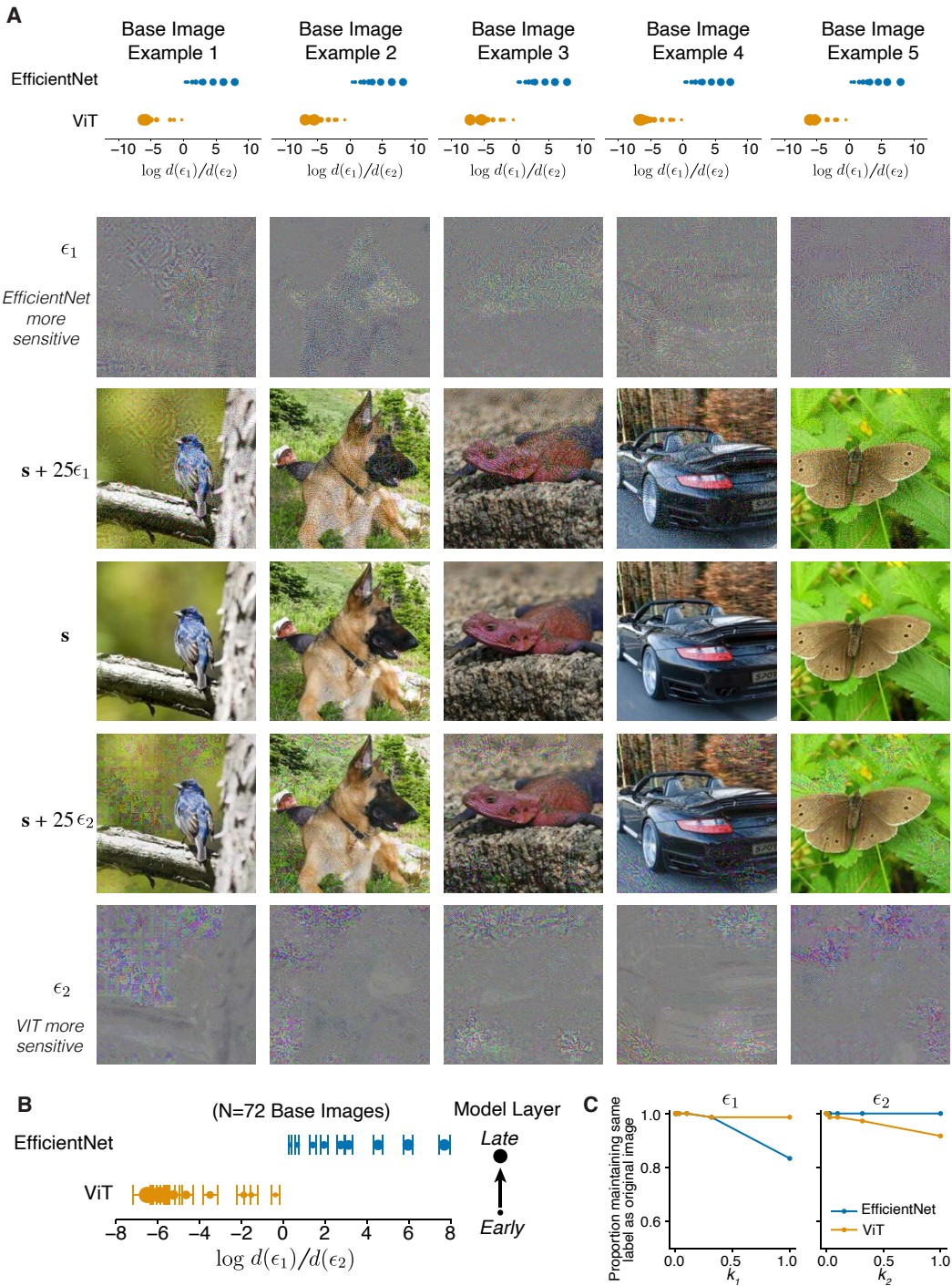

Figure SI.2: Comparison of ImageNet-1k trained EfficientNet and Vision Transformer. **A)** Example principal distortions and associated log sensitivity ratio plots when comparing layers of EfficientNet-B0 and ViT (Base-Patch16-224). **B)** When measured over 72 images from the ImageNet validation set (chosen from ImageNet-9, Xiao et al., 2021), the obtained principal distortions reliably separate the models by architecture—the layers of EfficentNet are more sensitive to distortion $\epsilon_1$, while the layers of ViT are more sensitive to $\epsilon_2$. The principal distortions are qualitatively different than those obtained when comparing AlexNet and ResNet50—for instance, the principal distortions $\epsilon_2$ (i.e., the distortion that the ViT layers are more sensitive) have notable grid artifacts corresponding to the size of the input patches to the ViT model. **C)** The distortions modify or preserve the label computed by each model, consistent with their predicted sensitivity (see Supp. Fig. SI.4 for a detailed explanation).

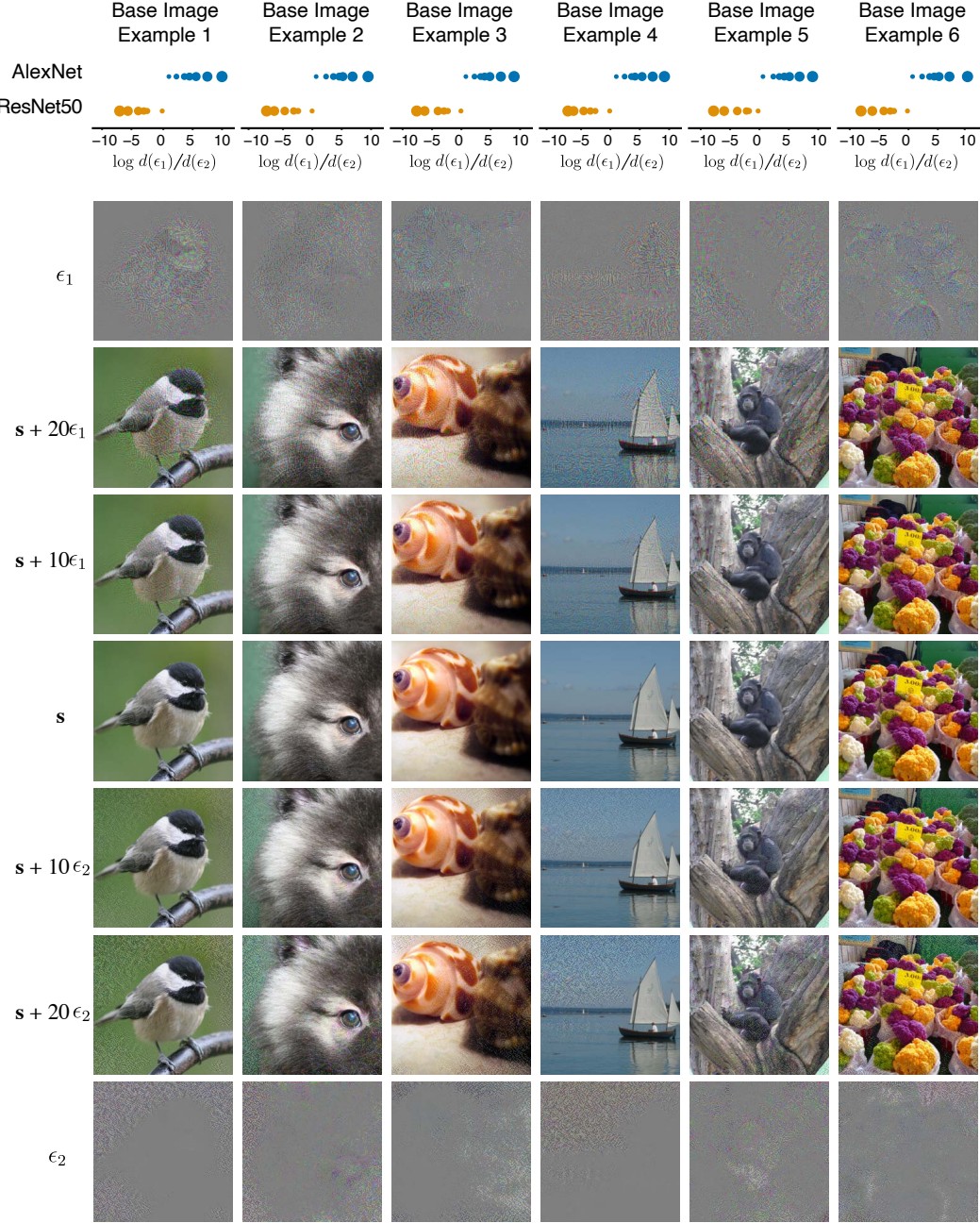

Figure SI.3: Principal distortions for ResNet50 and AlexNet for six example base images ($\mathbf{s}$, middle row). Each row shows base images with differently scaled additive perturbations of $\boldsymbol{\epsilon}_1$ and $\boldsymbol{\epsilon}_2$. Distortions $\{\boldsymbol{\epsilon}_1, \boldsymbol{\epsilon}_2\}$ are shown in isolation in the top/bottom rows, respectively. The log ratio plot for each image is at the top of the column.

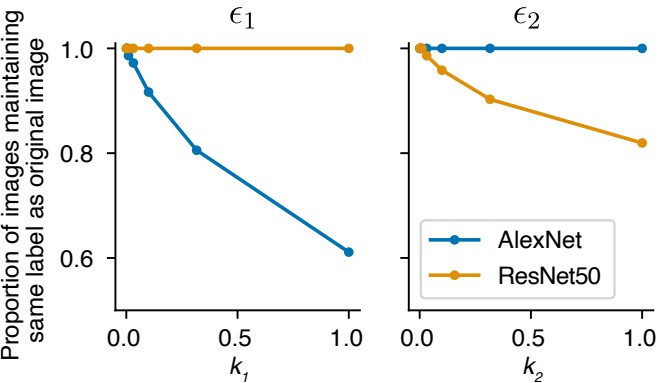

Figure SI.4: Adding a principal distortion to the base image changes the prediction of one DNN but not the other. Principal distortions were generated for intermediate layers of the AlexNet and ResNet architectures and were computed for a set of 72 base images from the ImageNet validation set (chosen from images included in ImageNet-9, Xiao et al., 2021). For each image $\mathbf{s}$ and principal distortion $\epsilon_i$, the distorted images $\mathbf{s} + k_i\epsilon_i$, with $k_i$ varying between 0 and 1, were classified by the DNNs used for the principal distotion generation (AlexNet or ResNet50). The left plot shows the proportion of images for which each DNN had the same classification for the distorted image $\mathbf{s} + k_1\epsilon_1$ as the base image $\mathbf{s}$, for $k_1$ varying between 0 and 1. The right plot is the same, but with $\epsilon_2$ and $k_2$ in place of $\epsilon_1$ and $k_1$.

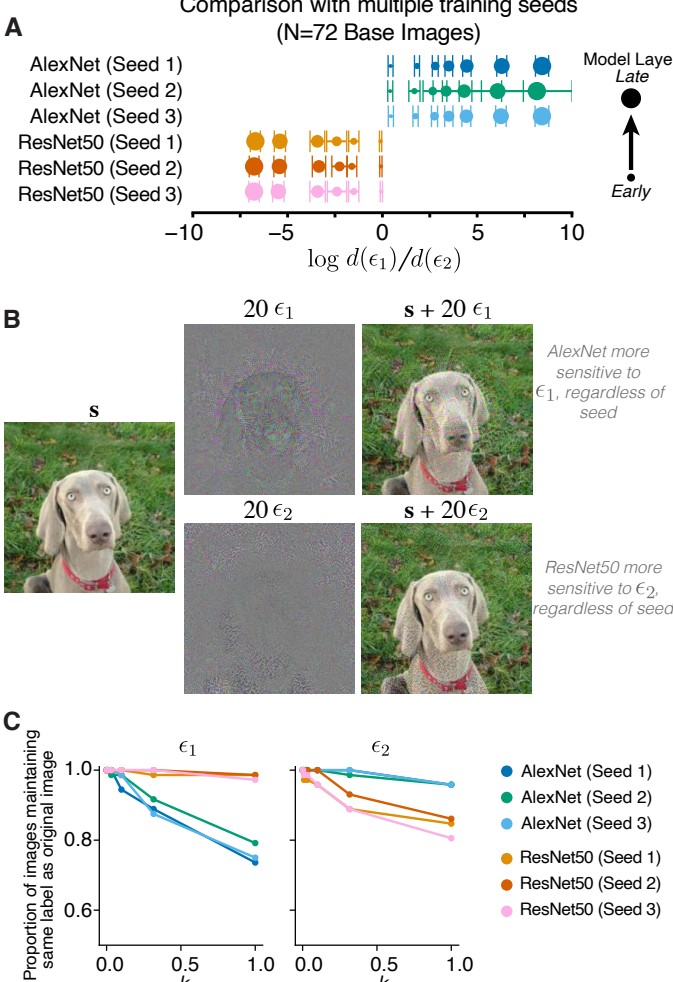

Figure SI.5: Principal distortions generated for multiple random initializations of AlexNet and ResNet50 consistently separate models by architecture type. **A)** Log sensitivity ratios of principal distortions for networks trained on standard ImageNet when comparing image representations at multiple layers of AlexNet and ResNet50, using three random initializations of each architecture. Principal distortions were computed for a set of 72 base images from the ImageNet validation set (chosen from images included in ImageNet-9, Xiao et al., 2021), and the mean ratio across these 72 images is plotted (error bars are standard deviation). **B)** Example base image, principal distortions, and distorted images. The principal distortions are qualitatively similar to those observed from a single architecture in Fig. 3. **C)** Proportion of images with a changed prediction class (compared to image $\mathbf{s}$) for the distorted images $\mathbf{s} + k_i\epsilon_i$. Results are similar to trends for single seeds of models from Supp. Fig. SI.4.

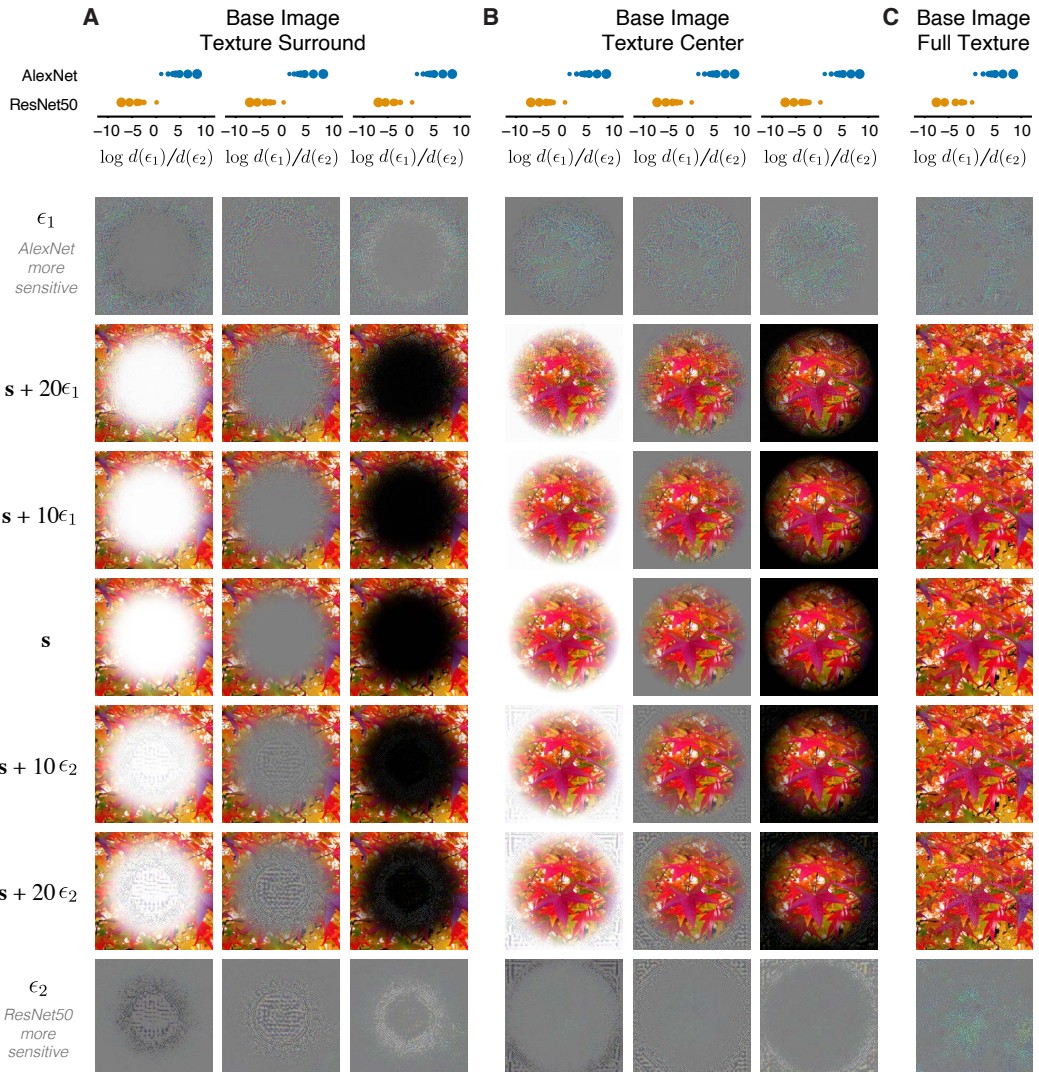

Figure SI.6: Principal distortions for ResNet50 and AlexNet for base images that have been constructed to have either (A) texture only in the periphery and a blank center (B) texture only in the center and a blank surround and (C) a full image of texture. The log ratio plot for each image is at the top of the column. Panels A and B highlight that the AlexNet architecture is more sensitive to the perturbation with power around the "stuff" of the image (i.e., the non-blank areas), while ResNet50 is more sensitive to a distortion focused on blank areas of the image, and these perturbations are not sensitive to the choice of low contrast ("black") or high luminance ("white") for the blank part of the space. Panel C highlights that in the case where there is not obvious blank area of the image, the perturbations become harder to interpret, and this reflects the locally adaptive property of the FI to the input image.

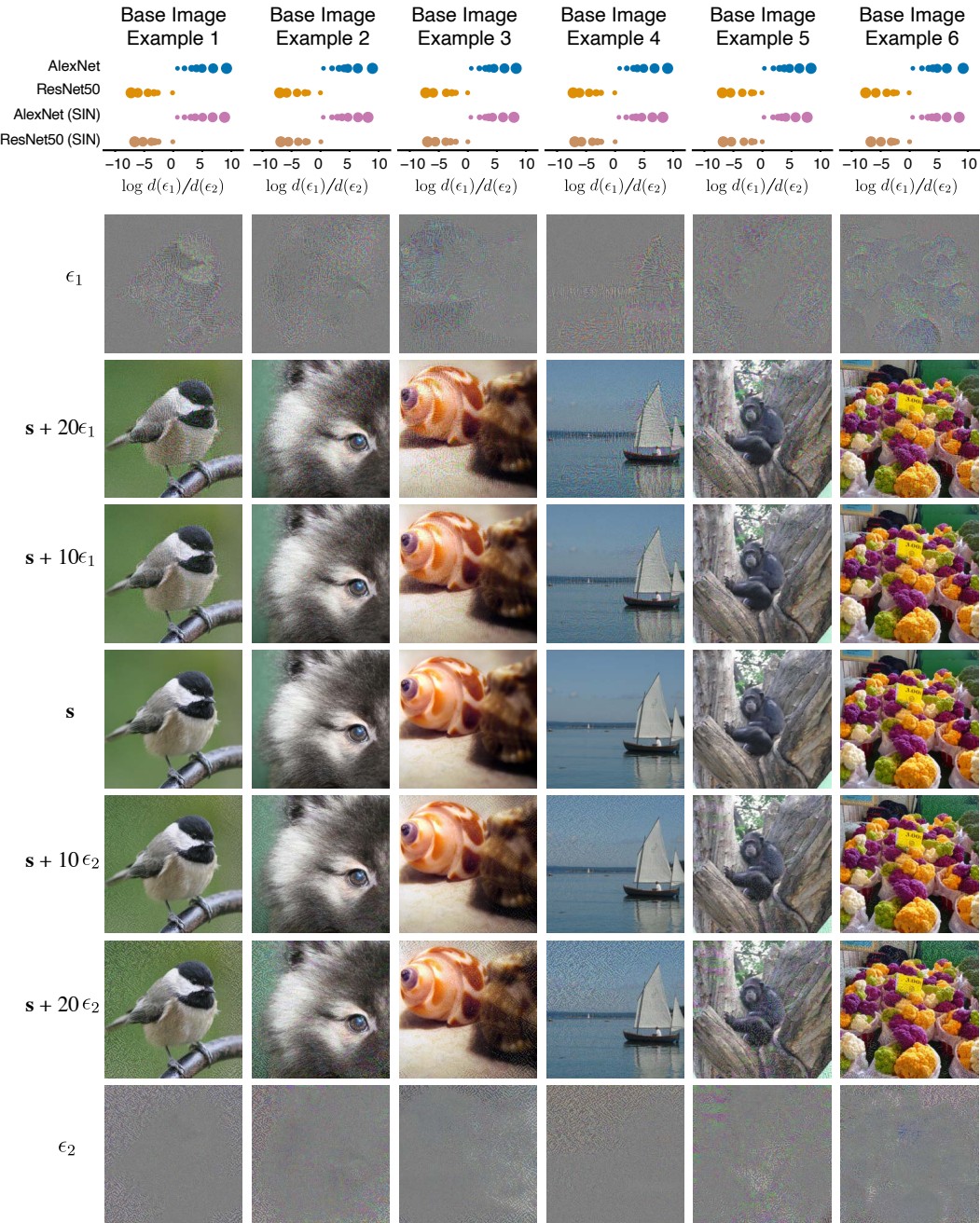

Figure SI.7: Principal distortions for standard ImageNet trained and Shape Image Net Trained (SIN) AlexNet and ResNet50 architectures for example base images.

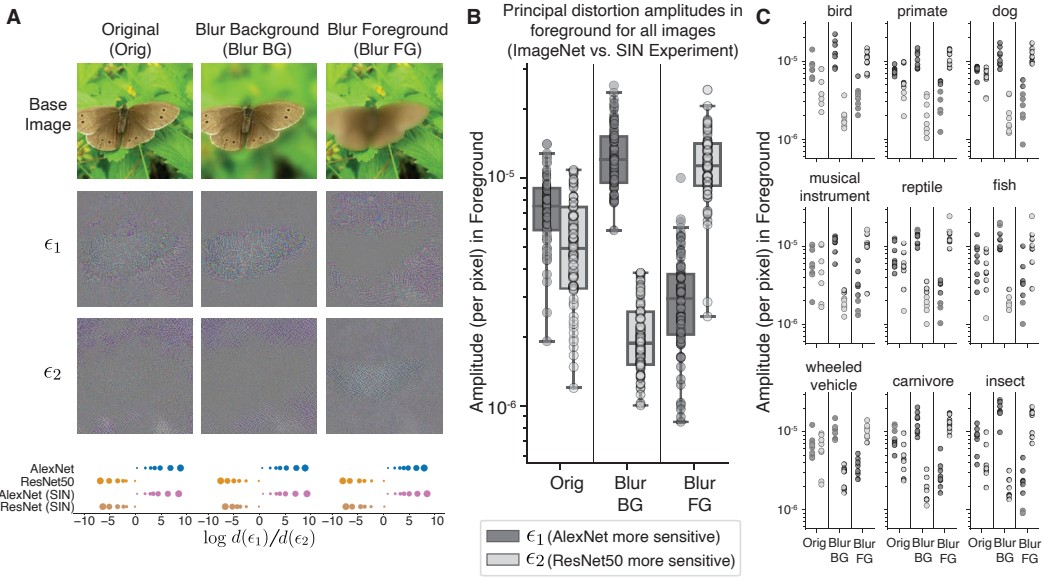

Figure SI.8: Control experiment dissociating foreground and background from high- and low-spatial frequency regions. **(A)** Example natural base image, image with a blurred background, and image with a blurred foreground. Images were chosen from ImageNet-9 (Xiao et al., 2021), which contains masks of the "foreground" parts of the image. Images are $224 \times 224$ pixels. To construct the blurred variants, a Gaussian filter with $\sigma = 5$ is was applied to the full image, and we used the provided image masks to construct the Blur Background (Blur BG) condition by adding the original image foreground to the blurred image background, and the Blur Foreground (Blur FG) condition by adding the original image background to the blurred image foreground. The associated optimal distortions are shown for each image, generated from the layers from AlexNet, ResNet50, SIN-Trained AlexNet, and SIN-Trained ResNet50. In the example image, the distortion AlexNet is most sensitive to ($\epsilon_1$) is biased towards higher frequency (non-blurred) parts of the image, while the distortion ResNet50 is most sensitive to ($\epsilon_2$) is biased towards low-frequency (blurred) parts of the image. We quantified this observation in **(B)** by masking the perturbation with the foreground mask, and measuring the average squared value of each perturbation in this region. If a perturbation is localized to the foreground, it will have a high perturbation magnitude on this plot. Each point on the plot corresponds to one of 72 randomly selected images from ImageNet-9 (Xiao et al., 2021), where 8 images were selected from each category, and the box and whisker plots are defined with the median given by the line across the box, the box extends from Q1 to Q3, and the whiskers extend to 1.5 IQR. Although there is a slight bias for distortion $\epsilon_1$ (defined as the perturbation AlexNet is more sensitive to), to have more power in the foreground of the image, this is likely due to the general biases of natural images where often times the background is blurred. If we blur the background, we see that this difference is exaggerated, while if we blur the foreground, the AlexNet perturbation shifts to be less concentrated in the foreground while the perturbation ResNet50 is more sensitive to ($\epsilon_2$) becomes more concentrated in the foreground. Points for individual categories are shown in **(C)**, which show the same trends, and results look similar when just including AlexNet and ResNet50 models in principal distortion generation (not shown).

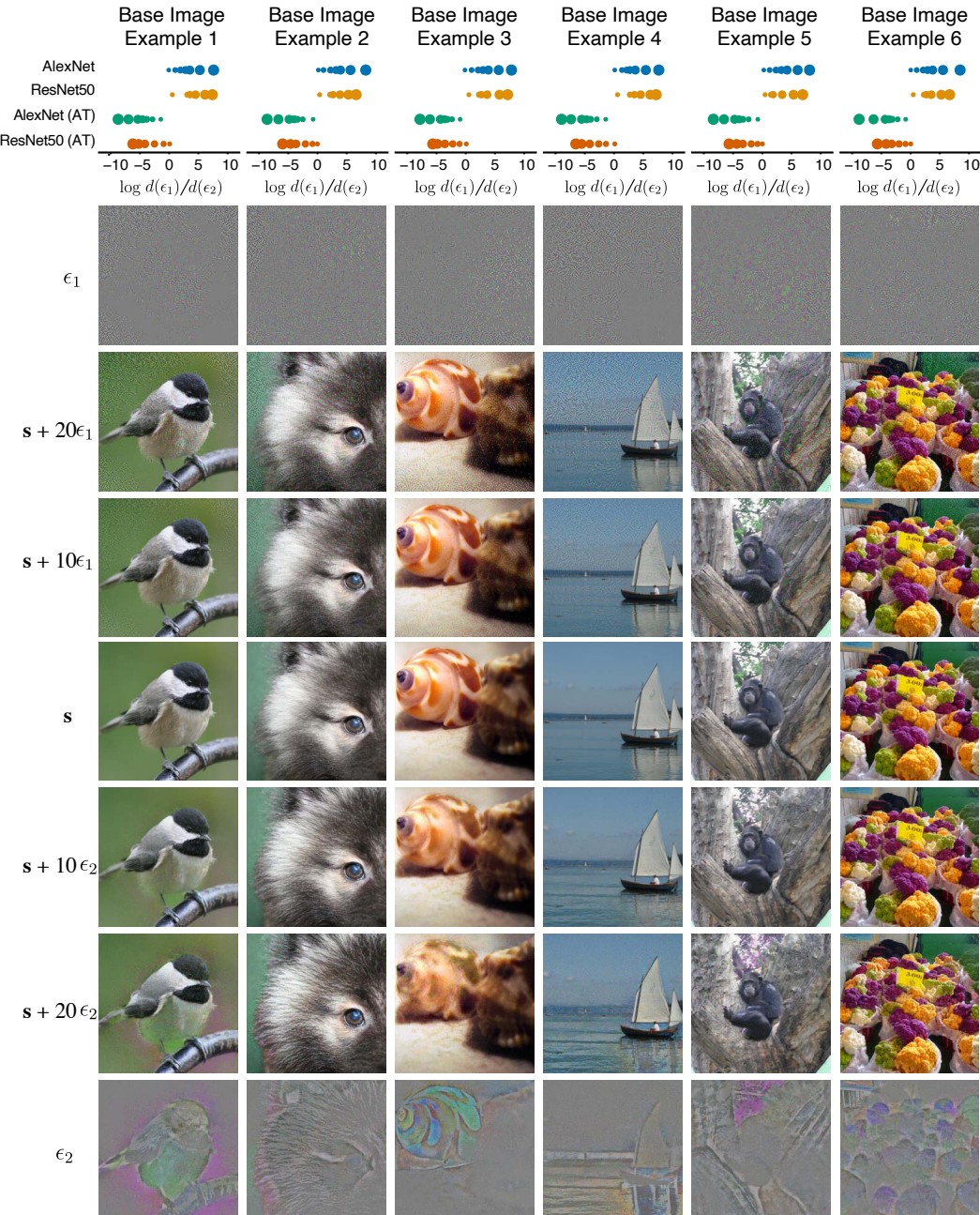

Figure SI.9: Principal distortions for standard ImageNet trained and adversarially trained AlexNet and ResNet50 architectures for example base images.

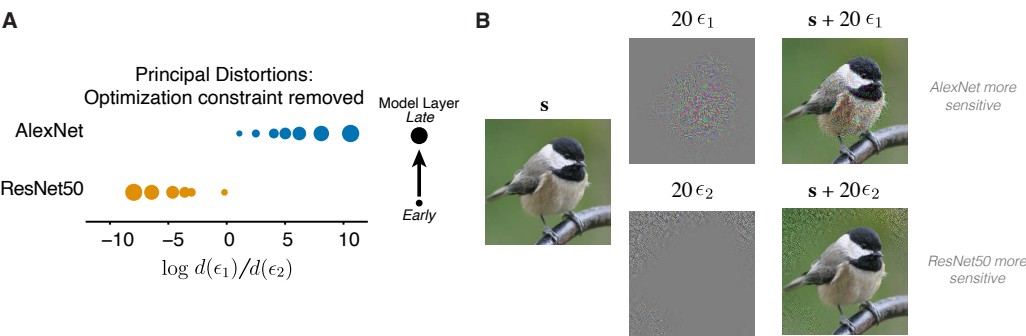

Figure SI.10: Example principal distortions for ResNet50 vs. AlexNet comparison where the pixel-wise min/max value for the perturbation has not been constrained to avoid clipping once the perturbation is scaled. Similar to the results in the main text, we see that the principal distortion to which AlexNet is more sensitive is focused on the part of the image with high-contrast texture features, while ResNet50 is more sensitive to the perturbation targeting relatively smooth parts of the image.

