# OpenReview forum: "Discriminating image representations with principal distortions"
_ICLR.cc/2025/Conference — ICLR 2025 Poster_

### Official Review · Reviewer_2NDt · 2024-11-03

**Soundness:** 4
**Presentation:** 3
**Contribution:** 3
**Rating:** 6
**Confidence:** 3

**Summary:**

The paper proposes a method to compare image representations by analyzing local geometries using the Fisher information matrix, enabling differentiation of models based on local sensitivity to distortions. For comparison, it has been applied to both early visual system models and deep neural networks, the method enables comparison of multiple models (>2) and can be used to compare model representations with human perception.

**Strengths:**

I found the paper's topic interesting and overall the language of the paper was clear. The paper targets an important problem in cognitive science and is a step toward understanding visual models better. The figures and qualitative results were sufficient to understand the concept and the differences. The paper is also based on a sound experimental framework to investigate the source of variability in results from both training data and architectural design.

**Weaknesses:**

I have a few questions and would like to know the authors' response to them:

1. (minor) It has been claimed in Figure 4 that the differences between AlexNet and ResNet come from the architecture than the training procedure. There is always randomness in the training where initialization of the model's weights and also the random split of data can effect one single model's performance. In fact, the model output may change if we train it on different seeds. I am wondering wouldn't that be a better approach to replicate the results on models trained on at least three different seeds? (basically 3 different versions of a model)

2. (minor) I expected to see a candidate from Transformers in the experiments, however, there was no candidate from those (currently) popular models. Can authors justify why they have not included any transformers such as ViT or Swin?

3. With Transformers, it has been a trend to show the attention maps (on tokens of an image) and show the behavior of the model. I am wondering if the authors can comment on this. Can the distortion maps be correlated to attention maps? This can ideally (if correlated) help with the explainability of attention-based models.

4. (major) Currently the manuscript is heavily populated with qualitative results. I strongly suggest the authors add a per-class sensitivity distribution for the dataset they are using for different models. This helps to understand on a population level and over different classes which models are more sensitive to high freq. localities and which ones are not.

5. I could not find a Dataset section in the manuscript :) Adding an independent dataset section would make it easier to understand the scale of experiments. currently, it is hard to find within the text

**Questions:**

mentioned everything in weaknesses

---

> ### Author Response · Authors · 2024-11-24
> **Author response to 2NDt**
>
> Thank you for your review and for highlighting the “sound experimental framework” of our paper. We appreciate the suggestion of further quantifying our results, and have added experiments to address this. See direct responses below.
>
> > (1) Re: Replicating results on models trained with multiple random seeds
>
> Yes, another way to validate the differences between the ResNet50 and AlexNet architectures would be to test models trained on different random seeds. The results with the Shape-Image-Net trained network seemed even stronger to us than the results that would be obtained with random seeds, because these models were trained with different datasets and behave differently with natural images. However, for completeness (and also because we had the same question), we ran an experiment with principal distortions for 6 simultaneous models where there are 3 seeds each for ResNet50 and AlexNet. This result is now included as **Supp. Fig. 14**.
>
>
> > (2) Re: Transformers
>
> We have run an experiment comparing an EfficientNet to a ViT and added this to **Supp. Fig. 12**. We are also constructing a Jupyter notebook similar to the early visual models, but that generates principal distortions from models in the Huggingface Pytorch Image Models repository (Timm), which we hope would encourage researchers to use our method and explore the rich and diverse set of models available to the community. Please see the general response **On the choice of AlexNet vs. ResNet50** for additional details.
>
> > (3) Re: Attention maps
>
> This is a very interesting question! It is true that our method and attention map visualization can both be used to interpret model representations. We have added a bit to the introduction and discussion highlighting that our approach is complementary to other techniques for model interpretability (see general response **Practical applications and model interpretability** for more details). That said, it would be very interesting in the future to run a similar analysis on a large set of transformers with a similar analysis as the foreground vs. background mask comparison (see below) but measure the distortion power weighted by the attention maps. But we leave this for future work.
>
> > (4) Re: Qualitative vs. Quantitative Results
>
> We have addressed this by adding a new experiment on images with blurred foregrounds and backgrounds, outlined in the general response. For this experiment, we explicitly chose to generate distortions on a dataset that has multiple images per class (compared to our original distortions, where we only had one image per class) so that we could also look at whether the results were impacted by class. The results are quite consistent regardless of the class, and so we believe this is not a class-specific effect but rather something about the image structure. For more details, see the general response section **DNN result quantification**.
>
> As another quantification, we looked at how our distortions generated for intermediate representations change the classification decisions of networks (see new **Supp. Fig. 13**, and also included in **Supp. Figs. 12C and 14C**. We found that the generated principal distortions drove changes in the final layer output, leading to classification changes with small image perturbations, and the fact that this occurs (even for perturbations generated for intermediate stages) strongly suggests that these distortions are “meaningful” in some way to the classification predictions. In future work, it would be interesting to generate distortions *specifically* for the classification layers of networks where one might see more differences between distortions generated from images of different classes.
>
> >(5) Re: Dataset section
>
> Thanks for the suggestion. We have added dataset sections to the methods describing the image datasets used to generate the principal distortions, one in the early visual model experiment (C.1) and one in the deep neural network experiments (D.1).

---

> > ### Comment · Reviewer_2NDt · 2024-11-27
> >
> > I appreciate the authors' thorough response and also adding the new experiments to their paper. My concerns have been addressed properly.

---

### Official Review · Reviewer_zmdX · 2024-11-03

**Soundness:** 4
**Presentation:** 4
**Contribution:** 3
**Rating:** 8
**Confidence:** 4

**Summary:**

The paper introduces a new method for comparing vision models that map input images to stochastic representations, leveraging the Fisher Information Matrix (FIM) to analyze the local geometry of image representations. This approach allows for the comparison of multiple models, including computational models of the visual system and deep neural networks (DNNs). By focusing on relative sensitivities to principal distortion directions in the stimulus (image) space, the method reveals differences between models that are not captured by global geometric approaches such as representational similarity analysis (RSA). The authors demonstrate the utility of their method through computational studies, showing consistent findings with previous research on early visual system models and uncovering differences in local image representations between models like AlexNet and ResNet50.

**Strengths:**

- The paper is clearly written and easy to follow, with a strong motivation that outlines how it builds on previous work.
- Presents a new method to compare the local geometry of image representations between models, generalizing prior approaches to accommodate multiple models.
- The method uncovers differences invisible to global geometric approaches, providing deeper insights into model behaviors.
- Includes several computational experiments that support the method's effectiveness and relevance.
- Discusses potential applications in studying biological vision, DNNs, and the interplay between them.
- Offers a well-rounded discussion that contextualizes the findings within the field.

**Weaknesses:**

- The technical advancement may be seen as an incremental improvement over existing metrics.
- The method primarily applies to stochastic representations, which are uncommon in DNNs; applying it to deterministic models may seem arbitrary.
- It is not entirely clear how the proposed local metric can be aggregated to provide global insights across different stimuli or datasets.

**Questions:**

1. How do you plan to extend this work to achieve more quantitative results? For instance, do you have ideas on quantifying the distinction between "noisy" versus "smooth" image distortions?
2. The authors essentially present a metric between models, which is based on the model’s representations for a single stimulus. It seems to me that the results and conclusions drawn from them could potentially disagree for different stimuli. How do you either choose a representative stimulus or how do you choose multiple and how could the results be aggregated?
3. How does your approach compare to, or how could it be integrated with, explainability methods based on saliency or relevance maps?
4. Could you clarify how you scaled the relative sensitivities between models of the early visual system (as mentioned on pages 5 and 6)? A revision of this explanation might enhance understanding.
5. Can you speculate on how the observed differences in sensitivity to noisy distortions between AlexNet and ResNet50 relate to their architectural differences or inductive biases?

Additional feedback:
- For deterministic representations in DNNs, the application of the FIM might seem less direct. It would be beneficial to elaborate on this aspect, perhaps by relating it to concepts like the pullback of the Euclidean metric onto the image space.
- Strengthening the discussion of experimental results and their key takeaways would enhance the paper. Specifically, clarifying how principal distortions inform us about a model's alignment with human visual representation, responses to adversarial examples, and contributions to interpretability would be valuable.
- Introducing more quantitative analyses or metrics could solidify the results and provide stronger evidence of the method's effectiveness.

---

> ### Author Response · Authors · 2024-11-24
> **Author response to zmdX**
>
> Thank you for your careful reading of our paper and for your thoughtful comments, questions and feedback. We have addressed each point below.
>
> > W1: Incremental progress over existing metrics
>
> If “metric” is meant in a precise sense (i.e. a notion of distance that is symmetric and obeys triangle inequality), we are unaware of any other practical metric for comparing the local geometry of two image representations. Alternatively, if “metric” is meant in the sense that there are other methods for measuring or comparing the local geometry of image representations, we agree that our work builds on existing methods (Berardino et al. NeurIPS 2017, Zhou et al. Unireps 2023). However, we believe that our extension is critical since it *defines a metric* (in the precise sense) so that given a collection of models, one can essentially perform a principled dimensionality reduction that allows for comparison of these models in a low-dimensional space (1-dimensional in all of our examples, but our approach can be readily generalized).
>
> > W2: Application of FIM to deterministic models
>
> This is a fair point, but we note several reasons why FIM may still be reasonable, beyond the fact that prior work such as Berardino et al. and Zhou et al. already provide precedent for our choice.
>
> First, when comparing deterministic models, we assume isotropic Gaussian noise, so the FIM is the squared Jacobian of the model, a natural object to analyze when computing the local sensitivity of a deterministic model to small input perturbations.
>
> Second, deterministic DNN models have become a prevailing model of the human visual system. It’s long been hypothesized that human perceptual discriminability is based on the ability of downstream brain regions to distinguish between stochastic representations in the visual system. Therefore, it seems necessary to have a unified framework for comparing deterministic models (e.g. DNNs) with human perception (which is based on stochastic representations).
>
> Third, many DNN models *are* stochastic (e.g. dropout layers, diffusion layers, VAE latents). Although we did not directly investigate these types of representations, our approach would apply naturally to these cases.
>
> > W3: Aggregation of metric across stimuli or datasets
>
> Indeed. When we first started, we thought that the ordering of the models, etc would strongly depend on the base image. It’s quite remarkable how consistent the orderings of the models is (e.g., see the error bars in Fig 3E). Therefore, even though each pair of principal distortions is only measuring the local geometry of the model at a specific point, they appear to be consistent across stimuli for the experiments presented in our paper. See more discussion in Q2 below.
>
> > Q1: Quantitative results
>
> Please see section **DNN result quantification** of our general response
>
> > Q2: Metric depends on the stimuli
>
> Absolutely. The “distance” between models is at a specific stimulus and there is no requirement that this distance be consistent across stimuli. The similarity we observe in the paper for different images might reflect architectural biases present in the whole image space. We expect there are other cases where models would behave differently for different stimuli, and in these cases additional experiments would need to be done for interpretation of the distortions. We are actively thinking about this, but leave direct suggestions for future work.
>
> > Q3: Interpretability methods
>
> Please see the section **Practical applications and model interpretability** in our general response.
>
> > Q4: Early visual models
>
> Please see the general response section **Clarification regarding early visual models** for a list of changes (additional text and figures) we made to clarify the section on early visual models.
>
> > Q5: Inductive biases
>
> We speculate that these differences in principal distortions are related to architectural differences. For example, we generated principal distortions for comparing layers of EfficientNet and a Vision Transformer (**Supp. Fig. 12**) and found that the principal distortions that the Vision Transformer is most sensitive to have notable grid artifacts that correspond to the size of the input patches to the model. Full experiments ablating architectural components like residual connections or changing the size of convolutional kernels would be interesting in the future.
>
> Additional Feedback:
> > FIM for deterministic models
>
> Yes, good point! For deterministic smooth models, the metric on stimulus space induced by the FIM is the pullback of the Euclidean metric in representation space. We added **Line 269**.
>
> > Strengthening discussion
>
> We think that the additions to the paper described in the general responses **DNN result quantification**, **Early visual model clarification** and **Practical applications and model interpretability** address this point.
>
> > Introducing quantitative analyses
>
> Please see **DNN result quantification** in the general response.

---

> > ### Comment · Reviewer_zmdX · 2024-11-25
> >
> > Thanks a lot for your thoughtful responses and adaptions to the manuscript. I would have no further questions from my side. I think my questions have been addressed. Please let me know should you have further questions.

---

### Official Review · Reviewer_s6Hs · 2024-11-04

**Soundness:** 3
**Presentation:** 3
**Contribution:** 3
**Rating:** 6
**Confidence:** 3

**Summary:**

This paper proposes a framework for comparing image representations regarding their local geometries through the Fisher information matrix by finding a pair of “principal distortions” that maximize the variance of the models under this metric. The experiments include (1) comparing a set of simple models of the early visual systems and 2) comparing a set of deep neural network models to reveal differences in the local geometry that arise due to architecture and training types.

**Strengths:**

1. The idea is simple but effective -- as the first approach to compare more than two models.
2. The problem statement and method description are well-written and clear to understand.
3. The experiment result is interesting, with meaningful discussions about texture bias and adversarial vulnerability. Visualizations are very helpful.
4. The supplementary is very informative.

.

**Weaknesses:**

1. It is not clear how this paper's principal distortions relate to human sensitivity. There seem to be no experiments to prove this statement, such as using human observers for evaluation of the principle distortions [1].
2. For example, in 4.1 Early Vision Models, it is not stated how to determine the effectiveness of this approach. There is no quantitative evaluation, and it is unclear how to understand the visualizations of the principle distortion images.
3. Are there some practical applications of this approach? Or how to use visualizations of principle distortions?
4. What kind of networks can be compared with this approach? Only networks for classifications? How about the models for detections, segmentations, and other applications?
5. This paper is over nine pages.

[1] Alexander Berardino, Johannes Balle, Valero Laparra, and Eero Simoncelli. Eigen-distortions of ´
hierarchical representations. Advances in Neural Information Processing Systems, 30:3531–3540,
2017.

**Questions:**

See weaknesses.

---

> ### Author Response · Authors · 2024-11-24
> **Author response to s6Hs**
>
> Thank you for your interest in our work, and in particular for being interested in how we could use the method to investigate human perception. We have included responses to each question below.
>
> >Q1 & Q2. Relationship to human sensitivity and effectiveness of the approach
>
> We agree that the early vision model results would be more complete with a full psychophysics evaluation of the distortion sets, however in the current paper we decided to prioritize highlighting how the method can be used in multiple different settings and with large number of models, rather than focus exclusively on models of human perceptual distortions as was done in (Berardino et al. NeurIPS 2017). Given that other reviewers have mostly highlighted the DNN parts of the paper and asked for more analysis/experiments with those, we believe that holding off on a large-scale psychophysics experiment on the early visual models is more appropriate for a separate paper (we would likely opt for a journal article, rather than a conference paper) where we could “do it right” (and more practically, it is unfortunately beyond our abilities to complete in the rebuttal period).
>
> That said, we clarified the early visual model section to make the connection to future experiments more explicit (see the general response section **Early Visual Model Clarification**).
>
> >3. Practical applications of principal distortions
>
> For early visual models, once again see general response **Early visual model clarification**. We have also added additional discussion about using the method for DNN interpretability, see general response **Practical applications and model interpretability**.
>
> >4. Generality of the approach (what other networks can be compared)
>
> Our method can be applied to any differentiable model. See the general response **On the choice of AlexNet & ResNet50** where we demonstrate the generality of our approach by applying it to more modern architectures. As the method investigates the local geometry of the image representation, it is not dependent on having a classification decision and is equally applicable to models trained for segmentation, detection, or even unsupervised methods. For instance, note that the parameters of the early visual models were optimized in Berandino et al. 2017 for predicting human distortion ratings (see more details in Supp. Section C, starting on line 854).
>
> > 5. Manuscript is over 9 pages
>
> This year, 10-page manuscripts were allowed for both the submission and rebuttal phases.

---

> > ### Comment · Reviewer_s6Hs · 2024-11-27
> >
> > Thanks for the authors' thorough response. My concerns have been solved.

---

### Official Review · Reviewer_7znR · 2024-11-05

**Soundness:** 3
**Presentation:** 3
**Contribution:** 2
**Rating:** 5
**Confidence:** 4

**Summary:**

The paper proposes a novel framework to compare local geometry within image representations, arguing that image representation encompasses both global and local geometry information. The authors introduce a method to quantify the local geometry of image representations using the Fisher information matrix and statistical techniques sensitive to local stimulus distortion. This approach aims to identify "principal distortions pairs," which maximize model variances and serve as optimal discriminative tools for evaluating model performance.

**Strengths:**

The framework's focus on comparing local geometry is innovative, and the use of the Fisher information matrix and sensitivity to local distortions represents a unique approach to quantifying image representation. This novel method to derive principal distortion pairs that maximize model variance offers a potentially valuable tool for model discrimination.

**Weaknesses:**

Model Selection: The authors demonstrate the metric’s functionality using older architectures, specifically AlexNet and ResNet. Given that AlexNet is largely obsolete in current practical applications, the paper would benefit from extending the evaluation to more contemporary and widely-used networks (e.g., EfficientNet, Vision Transformers). Demonstrating the framework's effectiveness across a variety of modern architectures would strengthen the claim that the metric is universally applicable and capable of distinguishing models.

Practical Utility and Generalizability: The manuscript does not clearly establish the practical relevance and utility of this framework in real-world applications. A more explicit discussion on the potential benefits of this metric in actual deployment scenarios, or in improving model interpretability and selection, would add significant value. Additionally, empirical results supporting the framework’s effectiveness across diverse, practical model architectures are essential to substantiate the generalizability and robustness of the proposed approach.

**Questions:**

Model Selection: The authors demonstrate the metric’s functionality using older architectures, specifically AlexNet and ResNet. Given that AlexNet is largely obsolete in current practical applications, the paper would benefit from extending the evaluation to more contemporary and widely-used networks (e.g., EfficientNet, Vision Transformers). Demonstrating the framework's effectiveness across a variety of modern architectures would strengthen the claim that the metric is universally applicable and capable of distinguishing models.

Practical Utility and Generalizability: The manuscript does not clearly establish the practical relevance and utility of this framework in real-world applications. A more explicit discussion on the potential benefits of this metric in actual deployment scenarios, or in improving model interpretability and selection, would add significant value. Additionally, empirical results supporting the framework’s effectiveness across diverse, practical model architectures are essential to substantiate the generalizability and robustness of the proposed approach.

---

> ### Author Response · Authors · 2024-11-24
> **Author response to 7znR**
>
> Thank you for your comments and suggestions. We think that the analyses based on your questions have significantly strengthened the paper. Direct responses are below.
>
> >Model Selection
>
> Thank you for the suggestion! We now compare two modern architectures (EfficientNet and Vision Transformers, **Supp. Fig. 12**). We are also constructing a Jupyter notebook that generates principal distortions from models in the Huggingface Pytorch Image Models repository (Timm), which we hope would encourage researchers to use our method and explore the rich and diverse set of models available to the community. For more details, see the general discussion section **On the choice of AlexNet & ResNet50**.
>
> >Practical Utility and Generalizability
>
> Please see our general discussion section **Practical applications and model interpretability** for a discussion on the practical utility of our method for neural network interpretability and a list of specific changes to the paper. We have also strengthened the paper with additional empirical results that demonstrate our method’s generality (see general response **DNN result quantification**). These results include quantification of our previous results on the locations of principal distortions of the ResNet50 and AlexNet architectures (**Supp Fig 15**), a demonstration that the perturbations affected the network classifications (**Supp Fig 13**), and (as noted above) an application of our method to modern architectures (general discussion section **On the choice of AlexNet & ResNet50**).

---

> ### Author Response · Authors · 2024-12-02
>
> Thank you again for taking the time to review our paper. We've demonstrated that our method generalizes to modern architectures (see **Supp Fig 12**) and elaborated on the practical relevance of our approach in the **Practical Applications and Model Interpretability** section of the general response. Please let us know if we have addressed your concerns.

---

### Official Review · Reviewer_WQar · 2024-11-08

**Soundness:** 3
**Presentation:** 4
**Contribution:** 3
**Rating:** 6
**Confidence:** 3

**Summary:**

This paper introduces a novel metric on image representations to measure differences in local geometry. The authors then leverage this metric to generate "principal distortions" that maximize variance across a set of models. Experimental results of the proposed method reveal qualitative differences in the local geometry of ResNet50 and AlexNet, which is quite interesting.

**Strengths:**

- This is well-written and organized paper; hence it is easy for readers to follow.

- The concept of exploring the local geometry of deep networks and examining the interplay between local geometry and the global structure of images is compelling.

- Developing a novel metric to compare two image representations is highly innovative.

**Weaknesses:**

- While the proposed method of using the Fisher Information Matrix to measure the sensitivity of a representation to a stimulus distortion seems reasonable, its effectiveness as a metric is unclear, or difficult to justify.

- Identifying the types of principal distortions to which the network is most sensitive is an interesting idea. However, the proposed method lacks a good validation plan to confirm the accuracy or reliability of these findings.

**Questions:**

- How is the stimulus-dependent function $f(s)$, defined after Eq. (1), computed in practice?
- If $I(s)$ is positive semi-definite, this seems to cause issues with the metric defined in Eq. (3) when it approaches zero.
- How are two image representations obtained from a single image $S$? Are $I_A(S)$ and $I_B(S)$ learned by different neural networks? Please clarify.
- How are the coefficients for $\epsilon_1$ and $\epsilon_2$ determined in the proposed approach?
- The interpretation and justification of the estimated principal distortions are difficult to justify. For instance, in Fig. 3, while the finding that AlexNet and ResNet are more sensitive to complementary parts of the images is interesting, its validity is unclear. Providing additional justification would strengthen the claims. One possible approach is to introduce adversarial noise to different parts of the images and evaluate its impact on downstream classification tasks, e.g., comparing the performance drop across different parts of the images.
- This reviewer finds it difficult to connect the identified distortions with the concept of local geometry. The method appears to focus more on differences in image intensities and textures. However, it is possible that this reviewer has misunderstood some aspects of the proposed approach. Please clarify.

---

> ### Author Response · Authors · 2024-11-24
> **Author response to WQar**
>
> Thank you for your comments, questions and suggestions. We have updated text and included some additional analyses to clarify the method and strengthen the paper, each detailed below.
>
> > W1: Effectiveness of the FIM as a metric.
>
> FIM has a long track record in computational neuroscience (e.g., see the references on **Lines 155-156**) and more recently in ML (e.g. Berardino et al. NeurIPS 2017, Zhao et al. “The adversarial attack and detection under the Fisher information metric” AAAI 2019). This is because it has several appealing properties such as providing a lower bound on the ability of an ideal observer to differentiate between two stimuli, which is especially relevant when comparing to human perception where discrimination thresholds are hypothesized to reflect the ability of downstream brain areas to distinguish between stochastic representations.
>
> > W2: Validation plan to confirm the accuracy or reliability of the findings.
>
> Please see the section **DNN result quantification** of our general response.
>
> Questions:
> > How is $f({\bf s})$ computed in practice?
>
> The function $f({\bf s})$ corresponds to any differentiable model; e.g., $f({\bf s})$ could correspond to the activation of a DNN layer in response to an input ${\bf s}$. We have edited the text on **lines 168-169** to further clarify this point.
>
> > What if ${\bf I}({\bf s})$ is degenerate?
>
> Yes, the metric is ill-defined if one of the FIMs is degenerate. A straightforward solution is to regularize the FIMs by adding the identity matrix scaled by a small constant. We clarify this point on **lines 961-964**.
>
> > How are two image representations obtained from a single image ${\bf s}$? Are ${\bf I}_A({\bf s})$ and ${\bf I}_B({\bf s})$ learned by different neural networks?
>
> In our experiments, each image representation corresponds to a model's response to a single image. For example, in the DNN experiments, a "model" corresponds to the activations in a layer of a DNN. For a full list of the "models" we used, see Sec. C (early visual models) and Sec. D (DNN layers) of the supplement. Note that we use pre-trained networks and the model weights remain fixed. We now emphasize this on **Line 352**.
>
> The FIMs ${\bf I}_A({\bf s})$ and ${\bf I}_B({\bf s})$ are *computed*, not *learned*. For example, for *deterministic* models $f_A({\bf s})$ and $f_B({\bf s})$, we assume isotropic Gaussian noise in which case the FIMs are ${\bf I}_A({\bf s})={\bf J}_A({\bf s})^\top{\bf J}_A({\bf s})$ and ${\bf I}_B({\bf s})={\bf J}_B({\bf s})^\top{\bf J}_B({\bf s})$, where ${\bf J}_A({\bf s})$ is the Jacobian of $f_A(\cdot)$ at ${\bf s}.$
>
>
> > How are the coefficients for $\boldsymbol{\epsilon}_1$ and $\boldsymbol{\epsilon}_2$ determined in the proposed approach?
>
> We assume you are referring to Figure 2 (if not, please let us know). Please see the general response **Early visual model clarification** and updated lines 315-322 of the text.
>
> > The interpretation and justification of the estimated principal distortions are difficult to justify.
>
> We now include quantification of our qualitative observations (see our general response **DNN result quantification**). Based on your suggestion, we tested how the addition of principal distortions to the base image changes the classification decision of each network (**Supp. Fig. 13**). We find that each generated principal distortion drove changes in the final layer output of one DNN (leading to classification changes with small image perturbations) but *not* the other DNN. This analysis raises very interesting future directions using the principal distortion method directly applied to classification outputs of networks, and potentially creating a form of adversarial example that maximally differentiates many classification outputs. Detailed analysis of principal distortions applied to classification layers is beyond the scope of the current paper which introduces the method, but would be quite interesting as a future direction. We appreciate your suggestion and think it opens up very interesting directions for future exploration!
>
> > Difficulty connecting the identified distortions with the concept of local geometry.
>
> The local geometry at a base image is closely related to how sensitive the model responses are to small distortions of the base image (see Sec. 2.1). Therefore, a model’s sensitivities to principal distortions capture aspects of the local geometry induced by the model. Once these principal distortions are computed, we focus on identifying properties that consistently characterize them. Ultimately, by characterizing the principal distortions, we hope to better understand the differences in local geometries of models (e.g. DNNs). Please let us know if this helps clarify our proposed approach. We have also added a line to the introduction **Lines 46-47** to help readers link local geometry to distortions.

---

> > ### Comment · Reviewer_WQar · 2024-12-03
> >
> > Thanks for the authors' responses. My questions are well clarified.

---

> ### Author Response · Authors · 2024-12-02
>
> Thank you once again for taking the time to review our paper. We've elaborated on our choice of the Fisher information matrix as a metric, as well as quantified the reliability of our findings. Please let us know if we have addressed your concerns.

---

### Author Response · Authors · 2024-11-24
**General Response**

Thank you for your thoughtful questions, comments, and suggestions. We are pleased that reviewers found our principal distortion method “innovative”, and drew connections to many different research areas. As a result of the reviewers' comments, we have performed several new analyses studying the principal distortions and included 5 new supplementary figures. Here, we address general questions raised by multiple reviewers. Specific responses to each reviewer are below.
# DNN result quantification
In our DNN experiments, we found that AlexNet is consistently more sensitive to distortions that are concentrated on the “stuff” in the image (portions of the image with higher variability) whereas ResNet50 is more sensitive to distortions that appear concentrated on relatively constant parts of the image. We now quantify this observation with a controlled experiment (**Supp Fig 15**) in which we generate principal distortions for the DNNs trained on ImageNet or Stylized Image Net after blurring the foreground or background of a base image. The principal distortion that ResNet50 is more sensitive to (regardless of training) is consistently concentrated on the blurred portions of the image (areas lacking high spatial frequencies, regardless of whether this is in the foreground or background), while AlexNet is more sensitive to the principal distortion that is concentrated on non-blurred parts of the image.

We also measured whether the principal distortions changed the network's classification decision (**Supp Fig 13**). We find that the distortions differentially change the classification decisions. The model that is more sensitive to distortion $\boldsymbol{\epsilon}_1$ changed its classification decision when $\boldsymbol{\epsilon}_1$ was added to the base image, while the other model did not change its classification decision. This points to future work that could investigate distortions explicitly generated to differentially change models’ classification decisions (we focused on intermediate representations).
# Early visual model clarification
There was some confusion about how we scaled the distortions before adding them to the base image in Fig 2C. We have modified the wording on **Lines 316-319** to clarify. We have also added **Supp Fig 11**, detailing how human perceptual experiments could be carried out to identify which model is best aligned with the human visual system.
# On the choice of AlexNet & ResNet50
Several reviewers asked why we did not generate principal distortions for modern DNN architectures. We opted to present controlled experiments on well-studied architectures with varied training procedures that were publicly available. Despite the fact that these networks have been well studied, the principal distortions revealed what appear to be novel and interpretable differences between the local sensitivities of AlexNet and ResNet50. We have made this choice more explicit on **Lines 357-360**.

That said, our method can be applied to any differentiable model. To directly demonstrate this with modern architectures, we generate principal distortions for layers of EfficientNet and a ViT (**Supp Fig 12**). We find that the distortions reliably differentiate the two networks and capture the hierarchical structure, are consistent across many base images, and are qualitatively distinct from the distortions found when comparing AlexNet vs ResNet50 (for instance, there are notable blocks corresponding to the patch size in the distortion that ViT is most sensitive to). We are also working on a Jupyter notebook that easily generates principal distortions from models in PyTorch Image Models (Timm), which we hope will encourage others to use principal distortions for model analysis.
# Practical applications and model interpretability
Multiple reviewers asked about practical applications of principal distortions and how they contribute to model interpretability. Principal distortions can be used as an exploratory analysis method for understanding how networks differ in terms of local sensitivity, which can then lead to further investigation and experimentation. This touches on many aspects of model interpretability. For example, principal distortions can be used to identify what a model is sensitive to relative to other models, while most other interpretability tools focus on single models. We added lines to the introduction (**Lines 53 and 72-74**) and discussion (**Lines 534-536**) to make this more explicit.

We also hope that the clarification about the early visual models demonstrates the practical applications in the domain of neuroscience/cognitive science. Overall, we think the examples in our paper highlight the broad applicability of our principal distortions method, and only scratch the surface of potential analyses and experiments to run with our approach. We are excited by the ways our method can be used for both neuroscience/cognitive science and in ML fields such as model interpretability.

---

### Author Response · Authors · 2024-12-03
**Thank you**

We'd like to express our gratitude to the reviewers for your thoughtful and detailed feedback on our submission. Your insights encouraged us to refine several aspects of our work, including clarifying the experiments on early visual models, quantifying and expanding the DNN results, and elaborating on the practical applications of our method, particularly highlighting its connection to model interpretability. Your constructive feedback significantly enhanced our submission and underscored the broad applicability of our approach. We are excited about the potential uses now highlighted in the discussion!

Following the discussion period, we were pleased that all of the reviewers who responded indicated that we successfully addressed all of their questions and concerns. Thank you again for your feedback and enthusiasm about our principal distortion method!

---

### Meta-Review · Area_Chair_aUrD · 2024-12-17

**Metareview:**

The current paper applies Fisher information w.r.t. multiple neural networks to find local distortions in the input space of image representations. The reviews highlighted the novelty and clear presentation. Most reviewers acknowledged that the authors' rebuttal has sufficiently addressed their concerns, and that the work is meaningful to extends such local metrics to multiple neural networks and useful for model interpretability.

In preparing the final version, please consider carefully the comments raised by all reviewers. Additionally, the authors mentioned that eq.(3) is a proper metric between positive semi-definite matrices, this is not true considering degenerate cases such as  $\epsilon'=\lambda \epsilon$, or $I_A=\lambda I_B$, or when $\epsilon$ or $\epsilon'$ lies in the kernel of $I_A$ or $I_B$ as the authors have mentioned in the rebuttal. Note a metric distance is strictly positive for any two different points. Please reword related sentences accordingly with proper citations and/or proofs.

**Additional Comments On Reviewer Discussion:**

4 out of 5 reviewers are positive and participated in the discussion. All the 4 reviewers have acknowledged that the authors have sufficiently addressed their concerns.

The other reviewer 7znR is not responsive after email correspondence. The weakness they raised have been addressed in the authors' revision.

There is mistake as eq.(3) is not a distance metric as the authors have claimed. It is mentioned in the meta-review for the authors' revision.

---

### Decision · Program_Chairs · 2025-01-22

Accept (Poster)